# Multiagent Reinforcement Learning in Games with an Iterated Dominance Solution

## Abstract

Multiagent reinforcement learning (MARL) attempts to optimize policies of intelligent agents interacting in the same environment. However, it may fail to converge to a Nash equilibrium in some games. We study independent MARL under the more demanding solution concept of iterated elimination of strictly dominated strategies. In dominance solvable games, if players iteratively eliminate strictly dominated strategies until no further strategies can be eliminated, we obtain a single strategy profile. We show that convergence to the iterated dominance solution is guaranteed for several reinforcement learning algorithms (for multiple independent learners). We illustrate an application of our results by studying mechanism design for principal-agent problems, where a principal wishes to incentivize agents to exert costly effort in a joint project when it can only observe whether the project succeeded, but not whether agents actually exerted effort. We show that MARL converges to the desired outcome if the rewards are designed so that exerting effort is the iterated dominance solution, but fails if it is merely a Nash equilibrium.

## 1 Introduction

Intelligent agents sharing a common environment are affected by the actions taken by their peers. Using reinforcement learning (RL) to derive agent policies becomes challenging since the environment becomes non-stationary for each agent when its peers adapt their behaviour through their learning process. One simple form of multiagent reinforcement learning (MARL) is independent learning, where each agent simply treats its experience as part of the non-stationary environment. Unfortunately, independent MARL fails to converge to a Nash equilibrium in many settings (Bowling, 2000; Shoham et al., 2003). To guarantee convergence to a Nash equilibrium, one must either examine restricted classes of games such as fully cooperative games (Claus & Boutilier, 1998; Bu et al., 2008; Panait et al., 2006; Matignon et al., 2007), or devise specialized algorithms that guarantee convergence (Hu & Wellman, 2003; Wang & Sandholm, 2003). We investigate independent MARL in games that are *solvable by iterated elimination of dominated strategies* (Moulin, 1979). We say that an action by an agent is dominated by another if the first action offers the agent a strictly lower reward than taking the second action, no matter which actions are taken by the other agents. In iterated elimination of dominated strategy we iteratively examine the actions of every agent, and remove strictly dominated actions, until no further actions can be removed. A game is *dominance solvable* if only one action profile survives the process of iteratively eliminating strictly dominated strategies.

We examine implications of the relation between iterated dominance and RL through applications in mechanism design, a field in economics that studies how to set incentives for rational agents, so as to achieve desired objectives. One key line of work in mechanism design deals with principal-agent problems (Holmstrom et al., 1979) holmstrom1982moral,grossman1992analysis,laffont2009theory, relating to a principal in charge of a joint project, whose success depends on the exertion of effort by multiple agents; the principal wishes to incentivize agents to maximally exert costly effort, but cannot observe how much effort any individual agent exerted.

**Our contribution:** We show that for dominance solvable games, multiagent reinforcement learners converge to the iterated dominance solution for simple and reasonable algorithms; in games with two actions per agent, REINFORCE (Williams, 1992) converges to the solution, and in games with more than two actions Monte-Carlo Policy Improvement (Sutton & Barto, 2018) converges when using importance weighted action value estimators. In contrast to a Nash equilibrium, which exists in

any game with a finite action set, not every game is dominance solvable. However, in mechanism design settings we *engineer* the game in order to achieve certain desired agent behaviors, and can thus *construct* games that are dominance solvable. We examine mechanism design to illustrate the applications of our work, empirically investigating a principal-agent problem. We show that an incentive scheme based on iterated dominance guarantees that independent reinforcement learners converge to the optimal solution for the principal, whereas under a scheme where exerting effort is only a Nash equilibrium, independent RL typically does **not** converge to an optimal solution.

## 1.1 PRELIMINARIES

**An $n$-player normal form game** is given by a set of players $I = \{a_1, \ldots, a_n\}$, and for each player $a_i$ a (finite) set of pure strategies $S_i$, and a utility function $u_i : S_1 \times S_2 \times \ldots \times S_n \to \mathbb{R}$, where $u_i(s_1, \ldots, s_n)$ denotes $a_i$'s utility when each player $a_j$ plays strategy $s_j$. For brevity, we denote the set of full strategy profiles $S = S_1 \times S_2 \times \ldots \times S_n$, and denote items in $S$ as $s \in S$ ($s = (s_1, \ldots, s_n)$, where $s_i \in S_i$). We also denote $S_{-i} = S_1 \times \ldots \times S_{i-1} \times S_{i+1} \times \ldots \times S_n$, and given a partial strategy profile $s_{-i} = (s_1, \ldots, s_{i-1}, s_{i+1}, \ldots, s_n) \in S_{-i}$ we denote $(s_{-i}, s_i) = (s_1, \ldots, s_{i-1}, s_i, s_{i+1}, \ldots s_n) \in S$. Given a normal form game $G$, we say agent $a_i$'s strategy $s_x \in S_i$ strongly dominates $s_y \in S_i$ if $a_i$'s utility is higher when using $s_x$ than when using $s_y$, no matter what strategies the other agents use, i.e.: agent $a_i$'s strategy $s_x \in S_i$ **strictly dominates** $s_y \in S_i$ if for any partial strategy profile $s_{-i} \in S_{-i}$ we have $u_i((s_{-i}, s_x)) > u_i((s_{-i}, s_y))$. We say player $a_i$'s strategy $s_x$ is $a_i$'s **dominant strategy** if it dominates all other strategies $s_i \in S_i$.

Game-theoretic solutions specify which outcomes are reasonable, under various assumptions of rationality and knowledge. We focus on a prominent procedure called iterated elimination of dominated strategies, and identify conditions under which learning agents converge to this solution. In cases where *every* agent has a dominant strategy, it seems reasonable to predict that each player would play their dominant strategy. Given a game $G$, we say a strategy profile $s = (s_1, \ldots, s_n) \in S$ is a **dominant strategy equilibrium** if for any agent $a_i$, strategy $s_i$ is a dominant strategy for $a_i$. However, in many games a player may not have a dominant strategy. A less demanding concept is that of a Nash equilibrium, which merely seeks a strategy profile where no player can improve their utility by unilaterally deviating. Given a game $G$ a strategy profile $s = (s_1, \ldots, s_n)$ is a **Nash equilibrium** if for any player $a_i$ and any alternative strategy $s_x \in S_i$ we have $u_i(s) \geq u_i(s_{-i}, s_x)$ (i.e. $u_i(s_1, \ldots, s_{i-1}, s_i, s_{i+1}, \ldots, s_n) \geq u_i(s_1, \ldots, s_{i-1}, s_x, s_{i+1}, \ldots, s_n)$). A mixed Nash equilibrium exists in games with finite strategy sets (Nash et al., 1950; Morgenstern & Von Neumann, 1953), but many games have multiple Nash equilibria, resulting in an *equilibrium selection* problem.

Another prominent concept is that of **iterated dominance** (Osborne & Rubinstein, 1994), where we iteratively remove dominated strategies, with eliminated strategies no longer having effect on future dominance relations. Given a game $G$ with players $I = \{a_1, \ldots, a_n\}$, strategy sets $S_1, \ldots, S_n$ and utilities $u_1, \ldots, u_n$, a (strict) domination elimination step $d$ is a triplet $d = (i \in I, s_l, \in S_i, s_h \in S_i)$, where the strategy $s_h$ strictly dominates $s_l$ for player $i$. The elimination step $d$ indicates that $s_l$ is eliminated from $G$ as it is dominated by $s_h$. Following the elimination step we get the game $G_d$, which is identical to $G$ except the strategy $s_l$ is removed from strategy set $S_i$ of player $i$ (i.e. the strategy set for $i$ in $G_d$ is $S_i \setminus \{s_l\}$), and the range of the utility function is restricted to this reduced strategy set). A dominance elimination sequence is a sequence $(G, d_1, G_{d_1}, d_2, G_{d_2}, \ldots, G_{d_{k-1}}, d_{k-1}, G_{d_k})$ where $G$ is an initial game and each $d_i$ is an elimination step from the game $G_i$ resulting in the game $G_{i+1}$. If no more dominance elimination steps can be taken from $G_k$, we say that the strategy profiles in $G_k$ *survive iterated elimination of (strictly) dominated strategies*. Further, if no more dominance elimination steps can be taken from $G_k$ and there is only one strategy remaining for each player, the game is called **(strict) dominance-solvable**. Iteratively eliminating dominated strategies is known to reserve Nash equilibria, and further when removing only strictly dominated strategies the procedure is "path-independent", yielding the same final strategy sets regardless of the order in which the dominated strategies were removed (Osborne & Rubinstein, 1994).

Our discussion focuses on normal-form game, but out results extend to **temporally extended** settings (games with multiple timesteps). We consider MARL in **Markov games** (Shapley, 1953; Littman, 1994), where in each state agents take actions (possibly given only partial observations of the true world state), with each agent obtaining an individual reward. We consider independent MARL, where agents each learn a behavior policy through their individual experiences interacting with one another in the environment. We discuss MARL in Markov games in Appendix 6.3, along experimental results.

One motivation for our work comes from **mechanism design**, a field of economics investigating how incentives should be set up so as to achieve desired outcomes in strategic settings where multiple agents interact. This was studied in settings ranging from government policy and social choice to auctions (Börgers, 2015; Nisan & Ronen, 2001; Krishna, 2009; Abdulkadiroğlu & Sönmez, 2003; Parkes & Singh, 2004). We focus on **principal-agent problems**, where agents take actions on behalf of another entity called the principal, but agents' interests may not align with the principal's (Holmstrom et al., 1979; Grossman & Hart, 1992; Laffont & Martimort, 2009). A key example is **efforts in a joint project** consisting of multiple tasks, each handled by an agent (Holmstrom et al., 1979; Holmstrom, 1982; Winter, 2004; Babaioff et al., 2006). We discuss this model in Section 4.

## 2 MULTI-AGENT RL AND DOMINANCE-SOLVABLE GAMES

We consider training multiple independent reinforcement learners in a game $G$ which is strict dominance-solvable. Each agent $i$ takes the role of player $i$ in the game $G$ and its possible actions are the strategies in $S_i$. Given the actions (strategy choices) of all agents we obtain a full strategy profile $s \in S_1 \times \ldots \times S_n$, and the reward each agent $i$ obtains is the respective payoff $u_i(s)$ in the game. As we consider training general RL agents in a domain that is a normal form game, we intermix game theoretic terminology (strategies and payoffs) and RL terminology (actions and rewards).

### 2.1 LEARNING DYNAMICS IN NORMAL FORM GAMES

Given the strategies $s_{-i} \in S_{-i}$ of all players except $i$, agent $i$ faces a single run of a game denoted as $b$ (reflecting the setting induced by the choices $s_{-i} \in S_{-i}$ of other players). The possible actions for agent $i$ are $S_i$, and any action $a \in S_i$ results in a reward $r_a^b = u_i(s_{-i}, a)$ as given by player $i$'s payoff in the game. However, agent $i$ simply selects an action and receives its obtained reward; it plays without ever gaining knowledge of which strategies were used by the other agents.

#### 2.1.1 REINFORCE AGENTS IN NORMAL FORM GAMES

We consider a REINFORCE (Williams, 1992) agent which maintains a score (logit) per each action, $x = x_1, \ldots, x_{m_i}$, and applies a softmax operation to transform these scores to the respective probabilities of choosing each action: $p_x(a) = \frac{\exp(x_a)}{\sum_{j=1}^{m_i} \exp(x_j)}$. Each agent starts with initial logits for $x_1, \ldots, x_n$. Fixing the choice $b$ of the other agents (relating to their chosen actions in $S_{-i}$), denote by $J^b$ the expected reward of the target agent, so $J^b = \sum_a p_x(a) r_a^b$. The exact **REINFORCE update** is: $x_{n+1} = x_n + \alpha \nabla_x J^b = \sum_a r_a^b \nabla_x p_x(a)$. As agents only take a single action each episode, this is typically estimated by substituting $\nabla_x J^b = \sum_a r_a^b \nabla_x p_x(a) = \sum_a r_a^b p_x(a) \nabla_x \log p_x(a) = E_{a \sim p_x} r_a^b \nabla_x \log p_x(a)$, then selecting a single action $a$ sampled from the distribution $p_x$ (parameterized by $x$). Given the softmax rule above for setting the action probability distribution $p_x$, and denoting Kronecker delta as $\delta_{ij}$ we have: $\frac{\partial p_x(i)}{\partial x_j} = p_x(i)(\delta_{ij} - p_j)$.

We examine MARL dynamics in dominance-solvable games, identifying conditions under which learning converges on the (strict) iterated dominance solution. Given the dominance elimination sequence $(G, d_1, G_{d_1}, d_2, G_{d_2}, \ldots, G_{d_{k-1}}, d_{k-1}, G_{d_k})$ (where the $d_i$s are elimination steps), one may hope the learning dynamics would "follow" the strategy elimination steps in the sequence, first lowering the probability on the dominated strategy of $d_1$ to (almost) zero, then lowering the probability on dominated strategy of $d_2$ and so on, until we remain with agents only playing the strategies of $G_{d_k}$. We show that this is indeed the case for MARL using REINFORCE when each agent has at most two strategies. For settings with an arbitrary number of actions per agents, we provide a similar proof for a variant of Monte-Carlo policy iteration given in Section 2.1.2.

#### 2.1.2 IMPORTANCE WEIGHTED MONTE-CARLO AGENTS IN NORMAL FORM GAMES

Monte-Carlo policy iteration (MCPI) is one of the simplest methods for control. It maintains an estimate of the expected reward for each strategy, updating the estimate after observing the outcome of every run of the game, and follows an $\epsilon$-greedy policy based on these estimates to guarantee exploration. To achieve convergence in dominance-solvable games, we use the specific estimator of Algorithm 1. At every step $t$, it maintains a score $x_i$ for every possible action $i \in S_i$. The scores are

softmaxed to derive a policy distribution $P$ over actions. We denote by $P_i$ the probability of choosing action $i \in S_i$. Every step, the agent selects an action $s_i$ from the current policy $P$, and depending on the actions $b \in S_{-i}$ taken by other agents, it receives a reward $r^b_{s_i}$. We denote the probability of selecting action $i$ under the policy $P$ at time $t$ as $P_{t,i}$ and the action taken by the agent at time $t$ as $A_t$. As an estimator for the reward when selecting action $i$, we use $\hat{R}_{t,i} = \frac{\mathbb{1}\{A_t=i\}r^b_{s_i}}{P_{t,i}}$ (in contrast to standard MCPI whose estimator is the average of past rewards when selecting action $i$). The score $x_i$ is increased by the estimator $\hat{R}_{t,i}$, and the scores $x$ are softmaxed to obtain an improved policy. As with MCPI, to maintain exploration we use an $\epsilon$-greedy version of this improved policy ( *in addition* to the exploration due to the softmax). Algorithm 1 is thus a variant of MCPI with an importance weighted reward estimator, which we study in the context of MARL in dominance-solvable games.

---

Algorithm 1: Importance Weighted Monte Carlo Policy Improvement (IW-MCPI), for agent $l$

---

1: **procedure** IMPORTANCE WEIGHTED MCPI
2:   $x_i \leftarrow 0$ (for all $s_i \in S_l$)
3:   **for** $t = 1, 2, \ldots$ **do**
4:    For each $i$ let $Q_i = \frac{\exp(x_i)}{\sum_{j=1}^k \exp(x_j)}$ // Compute softmax of $x$
5:    For each $i$ let $P_i = (1 - \epsilon_t)Q_i + \epsilon_t/k$ // Compute $\epsilon_t$-greedy policy derived from $Q$
6:    Sample action $A_t \sim P$
7:    Other agents select (unknown and unobserved) strategies $b \in S_{-l}$
8:    Play $A_t$ in the game, obtaining reward $r^b_{A_t}$
9:    $\hat{R}_{t,i} \leftarrow \frac{\mathbb{1}\{A_t=i\}r^b_{A_t}}{P_i}$
10:    $x_i \leftarrow x_i + \hat{R}_{t,i}$
11:   **end for**
12: **end procedure**

---

## 3   CONVERGENCE OF RL TO AN ITERATED DOMINANCE SOLUTION

We show that MARL in dominance-solvable games converges to the iterated dominance solution using the above MCPI (Algorithm 1), or under REINFORCE in the two action case. One may consider two training modes. In the *serial mode*, we cycle through the agents, each time performing RL updates for the policy of the current agent *while holding the policies of other agents fixed* for many iterations (enough for the policy to converge and eliminating a strategy). As we fix the strategies of others when training each agent, the process "follows" the domination elimination sequence. Another training mode is a *parallel mode*, where we update the policies of *all agents* following the experience gained in each episode (Littman, 1994). Our convergence results hold for both modes, but handling the parallel mode requires the more intricate conditional expectation analysis of Theorem 3.4.

### 3.1   BINARY ACTION CASE

Consider a dominance-solvable game and MARL using REINFORCE. As discussed in Section 2, given the strategic choices of other agents $b = s_{-i} \in S_{-i}$, agent $i$ faces a run of the game, with reward $r^b_a = u_i(s_{-i}, a)$ depending on $i$'s action $a$ and the strategies $b$ of the other agents. Each agent $i$ performs the REINFORCE updates of Section 2.1.1 based only on its action $a$ and obtained reward $r^b_a$, without ever becoming aware of the strategies $b = s_{-i}$ taken by others. A dominance elimination step $d = (i, s_l, s_h)$ includes a dominated strategy $s_l$ and dominating strategy $s_h$ for agent $i$, where $s_h$ strictly dominates $s_l$, so no matter what strategies $s_{-i}$ other players choose, player $i$ obtains a strictly greater utility from $s_h$ than $s_l$; Thus for any $s_{-i}$, $u_i(s_h, s_{-i}) > u_i(s_l, s_{-i})$, or in other words, for any setting $b \in S_{-i}$ that agent $i$ may be playing, action $s_h$ has a higher payoff than $s_l$, so $r^b_{s_h} > r^b_{s_h}$.

**Lemma 3.1.** *Let $B$ be a set of settings with two actions $s_l$ and $s_h$, where for any setting $b \in B$ the respective rewards for $s_h$ is strictly higher than for $s_l$, so $r^b_{s_h} > r^b_{s_l}$. The REINFORCE update eventually places a mass as close to zero as desired on the dominated action $s_l$.*

*Proof.* We consider applying the update for setting $b$, and a baseline of the lower reward $r^b_{s_l}$. Denote the minimal gap between the rewards of the actions as $g = \min_{b \in B}(r^b_{s_h} - r^b_{s_h}) > 0$. We have:

$\nabla_x J^b = \sum_a r_a^b \nabla_x p_x(a) = r_{s_h}^b \nabla_x p_x(s_h) + r_{s_l}^b \nabla_x p_x(s_l)$. The baseline argument for policy gradient updates states that for any constant $c$ we have: $\nabla_x(\sum_a p_x(a) r_a) = \nabla_x(\sum_a p_x(a)(r_a - c))$. Using the baseline argument with $c = r_{s_l}^b$ we get: $\nabla_x J^b = (r_{s_h}^b - r_{s_l}^b) \nabla_x p_x(s_h) + (r_{s_l}^b - r_{s_l}^b) \nabla_x p_x(r_{s_l}^b) = (r_{s_h}^b - r_{s_l}^b) \nabla_x p_x(s_h) \geq g \nabla_x p_x(s_h)$. Thus, regardless of the setting $b$ used, the update increases the probability of the dominating action, at least as much as the update for the minimal gap setting does. Repeatedly applying the update for the minimal gap setting eventually places negligible probability on the dominated action, so this is also the case for any update sequence (of any of the settings). $\quad\square$

**Theorem 3.2.** *Let $G$ be a dominance-solvable game which has a single strategy profile $s \in S_1 \times \ldots \times S_n$ surviving iterated elimination of strictly dominated strategies, and where every player has at most 2 strategies, and consider agents trained independently using the REINFORCE update. Then the agents converge to the iterated elimination solution $s$.*

*Proof.* Consider an iterated elimination sequence $(G, d_1, G_{d_1}, d_2, G_{d_2}, \ldots, G_{d_{k-1}}, d_{k-1}, G_{d_k})$. The first elimination $d_1 = (i, s_l^1, s_h^1)$ relates to agent $i$ who faces different settings due to other agents playing different strategies, but whose payoff under some $s_h \in S_i$ strictly dominates $s_l \in S_i$. Lemma 3.1 shows it eventually places negligible mass $\epsilon_1$ on the dominated action (for as low $\epsilon_1$ as desired). We examine the second elimination step $d_2 = (j, s_l^2, s_h^2)$. While in the original game $j$ has faced some settings $b' \in S_{-j}$ where $s_l^2 \in S_j$ got a higher reward than $s_h^2 \in S_j$, these settings are encountered less and less frequently. Consider a target probability $\epsilon_2$ for agent $j$ to select the dominated strategy $s_l^2$. By Lemma 3.1, there is a number $k$ of steps where if we train agent $j$ for $k$ steps only on settings where $s_l^2$ is dominated by $s_h^2$, $j$ places a mass of at most $\epsilon_2$ on $s_l^2$. By the union bound, the probability of encountering a "wrong" setting (with $s_l^2$ not dominated by $s_h^2$) is at most $k\epsilon_1$; as $\epsilon_1$ is as small as desired, the probability of agent $j$ not reaching the target (a mass of at most $\epsilon_2$ on $s_l^2$) is also as small as desired. Applying this argument over the elimination sequence, we conclude that agents converge on the single strategy profile $s$ surviving iterated elimination. $\quad\square$

Our proof of Theorem 3.1 iteratively applies Lemma 3.1, which holds when players have at most two strategies. Section 3.2 provides similar results to Theorem 3.2 for more than two actions, but under the MCPI variant of Algorithm 1. Section 4 backs the theory up through experiments.

## 3.2 Convergence in Dominance-Solvable Games for Importance Weighted MC

We consider agents using Algorithm 1 (IW-MCPI), and show that when an action $i$ dominates action $j$, IW-MCPI eventually stops choosing the dominated action. We assume rewards are normalized to the range $[0, 1]$. Denote the IW-MCPI estimator for the reward of action $i$ in time $t$ as $\hat{R}_{t,i}$, where $\hat{R}_{t,i} = \frac{\mathbb{1}\{A_t = i\} r_{A_t}^b}{P_{t,i}}$ ($r_{A_t}^b$ depends on the agent's action $A_t$, and the actions $b$ taken by others). The reward estimators $\hat{R}_t = (\hat{R}_{t,1}, \hat{R}_{t,2}, \ldots, \hat{R}_{t,k})$ are then converted to scores per action where $S_{t,i} = \sum_{j=1}^{t} \hat{R}_{j,i}$, and the scores are converted to a distribution $Q_t = (Q_{t,1}, Q_{t,2}, \ldots, Q_{t,k})$ by taking the softmax: $Q_t = Softmax(S_t)$. $Q$ encodes a "greedy" policy, which is then converted to an $\epsilon_t$-greedy policy $P_t$: $P_{t,i} = \frac{\epsilon_t}{k} + (1 - \epsilon) Q_t(i)$. We anneal the value of $\epsilon_t$ towards zero over time. Note that over time $t$, the scores $S_{t,i}$ are a sequence of random variables $S_{1,i}, S_{2,i}, \ldots, S_{\tau,i}$ where each $S_{t,i}$ is dependent on the earlier variables $S_{1,i}, \ldots, S_{t-1,i}$. We denote the conditional expectation of $S_{t,i}$ given the previous variables as: $\mathbb{E}_t(S_{t,i}) \triangleq \mathbb{E}[S_{t,i} | S_{1,i}, S_{2,i}, \ldots, S_{t-1,i}]$. In other words, $\mathbb{E}_t$ denotes the conditional expectation with respect to the observations by player $i$ at the start of round $t$.

**Theorem 3.3.** *Assume that $\epsilon_n$ is non-increasing and $\lim_{n \to \infty} \epsilon_n = 0$ and $\lim_{n \to \infty} \frac{\log(n)}{n^2} \sum_{t=1}^{n} \frac{1}{\epsilon_t} = 0$. Fix an agent and let $\bar{R}_{t,i}$ be the expected reward for the agent when playing action $i$ in round $t$. Then the following holds with probability one: $\lim_{n \to \infty} \frac{1}{n} \sum_{t=1}^{n} \left| \hat{R}_{t,i} - \bar{R}_{t,i} \right| = 0$.*

*Proof.* $\mathbb{E}_t[\hat{R}_{t,i}] = \bar{R}_{t,i}$ and $\mathbb{E}_t[\hat{R}_{t,i}^2] = \frac{\bar{R}_{t,i}^2}{P_{t,i}} \leq \frac{k}{\epsilon_t}$. Let $X_t = \hat{R}_{t,i} - \bar{R}_{t,i}$. Freedman's inequality (Freedman et al., 1975) states that for a sequence of random variables $X_1, \ldots, X_t$ (each depending on the previous ones in the sequence), with high probability of at least $1 - \delta$ the following holds:

$|\sum_{t=1}^{n}(X_t - \mathbb{E}_t[X_t])| \le c\sqrt{\sum_{t=1}^{n} Var_t[X_t] \log \frac{1}{\delta}}$. Applying Freedman's inequality we obtain:

$$P\left(\left|\sum_{t=1}^{n} X_t\right| \ge \sqrt{2k \sum_{t=1}^{n} \frac{1}{\epsilon_t} \log\left(\frac{2}{\delta}\right)} + \frac{3k}{2\epsilon_n} \log\left(\frac{2}{\delta}\right)\right) \le \delta.$$

Then by a union bound it follows that with probability at least $1 - \delta$ the following holds for all $n$:

$$\left|\sum_{t=1}^{n} X_t\right| \le \sqrt{2k \sum_{t=1}^{n} \frac{1}{\epsilon_t} \log\left(\frac{2n(n+1)}{\delta}\right)} + \frac{3k}{2\epsilon_n} \log\left(\frac{2n(n+1)}{\delta}\right). \tag{1}$$

The assumptions that $\epsilon_n$ is nonincreasing and $\lim_{n\to\infty} \frac{\log(n)}{n^2} \sum_{t=1}^{n} \frac{1}{\epsilon_t} = 0$ imply that $\lim_{n\to\infty} \frac{\log(n)}{n\epsilon_n} \le \lim_{n\to\infty} \frac{\log(n)}{n^2} \sum_{t=1}^{2n} \frac{1}{\epsilon_n} = 0$. Combining this with (1) shows that with probability one we have: $\lim_{n\to\infty} \frac{1}{n} |\sum_{t=1}^{n} X_t| = 0$ ☐

**Theorem 3.4.** *Let $G$ be a dominance-solvable game which has a single strategy profile $s \in S_1 \times \ldots \times S_n$ surviving iterated elimination of strictly dominated strategies. Consider agents trained independently using Algorithm 1. Provided that $\lim_{n\to\infty} \frac{\log(n)}{n^2} \sum_{t=1}^{n} \frac{1}{\epsilon_t} = 0$, the agents converge to the iterated elimination solution $s$.*

*Proof.* By setting $\epsilon_t = 1/t^p$, for any $p \in (0, 1)$, the assumptions on $\epsilon_t$ ensure that we can apply Theorem 3.3 (for all $i$, with probability 1). We show players' strategies converges to the iterated dominant profile. Suppose there exists a round $\tau_1$ after which action $i$ is dominated by action $j$, which means there exists a $g > 0$ such that for all $t \ge \tau_1$ it holds that $\bar{R}_{t,i} \le \bar{R}_{t,j} - g$. Then:

$$\frac{1}{n}\sum_{t=1}^{n} \hat{R}_{t,j} - \hat{R}_{t,i} = \frac{1}{n}\sum_{t=1}^{n}(\hat{R}_{t,j} - \bar{R}_{t,j}) + \frac{1}{n}\sum_{t=1}^{n}(\bar{R}_{t,i} - \hat{R}_{t,i}) + \frac{1}{n}\sum_{t=1}^{n}(\bar{R}_{t,j} - \bar{R}_{t,i})$$

$$\ge \frac{1}{n}\sum_{t=1}^{n}(\hat{R}_{t,j} - \bar{R}_{t,j}) + \frac{1}{n}\sum_{t=1}^{n}(\bar{R}_{t,i} - \hat{R}_{t,i}) + \frac{\tau_1 - 1}{n} + \frac{g(n - \tau_1)}{n}. \tag{2}$$

Taking the limit as $n$ tends to infinity shows there exists a time $\tau_2$ such that for all $n \ge \tau_2$ we have $\sum_{t=1}^{n} \hat{R}_{t,j} - \hat{R}_{t,i} \ge ng/2$. Therefore for any $n \ge \tau_2$ we have:

$$Q_{n,i} = \frac{\exp\left(\sum_{t=1}^{n-1} \hat{R}_{t,i}\right)}{\sum_{l=1}^{k} \exp\left(\sum_{t=1}^{n-1} \hat{R}_{t,l}\right)} \le \frac{\exp\left(\sum_{t=1}^{n-1} \hat{R}_{t,i}\right)}{\exp\left(\sum_{t=1}^{n-1} \hat{R}_{t,j}\right)} \le \exp\left(-ng/2\right). \tag{3}$$

Hence, $P_{n,i} = (1 - \epsilon_n)Q_{n,i} + \epsilon_n/k \le \epsilon_n + \exp(-ng/2)$. Since $\lim_{n\to\infty} \epsilon_n = 0$ by assumption, it follows that $\lim_{n\to\infty} P_{n,i} = 0$ almost surely. The previous part shows that if action $i$ is dominated after some round $\tau_1$, then for any $\epsilon > 0$ there exists a round $\tau_3$ such that $P_{n,i} \le \epsilon$. Choosing $\epsilon$ sufficiently small and iterating the argument completes the proof in the same way as Theorem 3.2. ☐

## 4 EMPIRICAL ANALYSIS OF PRINCIPAL-AGENT GAMES

**Our environment** is a simulation of a prominent problem studied by economists, called the *principal agent problem* (Holmstrom et al., 1979; Holmstrom, 1982; Winter, 2004; Babaioff et al., 2006), through which we show how our results can be used to design mechanisms for reinforcement learners. It considers a project which requires completing multiple tasks, each handled by an agent. Normally each task succeeds with a low (but non-zero) probability, which increases when the handling agent exerts additional effort. The project succeeds only if all its tasks succeed, in which case the principal stands to gain a large monetary amount (Appendix 6.2.1 considers a model where some task failures are allowed). The principal thus wants to make sure as many agents as possible exert effort. A dilemma arises when exerting effort is costly for the agents (i.e. incurs an immediate negative reward); A natural way to compensate for that is for the principal to offer agents a reward based on the effort they exerted. However, in principal-agent settings, the principal only knows whether the *entire project* succeeded, and is incapable of observing whether any individual agent exerted

effort (note tasks succeed with a non-zero probability even without effort). Thus, it can only promise each agent $i$ a reward $r_i$ offered only when the entire project is successful. We refer to the promised rewards $r = (r_1, \ldots, r_n)$ as a reward scheme. Each such reward scheme induces a game played by the agents, and the principal gets to *design the game*, by selecting the reward scheme. On the one hand, the higher the rewards, the more incentive agents have to exert effort. On the other hand, the rewards are costly to the principal, so they want to minimize them. One possible reward scheme is a **Nash equilibrium implementation**, where the principal sets rewards so that the profile where all agents exert effort is a Nash equilibrium (Babaioff et al., 2006). A Nash scheme may seem tempting to the principal as it offers low rewards. However, independent MARL may not converge to a Nash equilibrium (Lanctot et al., 2017), and there may be multiple equilibria, so agents may converge on an undesired equilibrium. A scheme at the other end of the scale is a **dominant strategy scheme**, where the principal promises each agent a reward high enough to make exerting effort a dominant strategy, so each agent would rather exert effort no matter what others do. Under this scheme MARL converges on exerting effort, but it is expensive to the principal. We show that an **iterated dominance scheme** is a good middle ground, guaranteeing convergence to the desired equilibrium at a far cheaper cost.

**Environment parameters:** we simulate a project which depends on five tasks $T = \{t_i\}_{i=1}^{n=5}$, a cost $c = 10$ for exerting effort, and where any task $t_i$ succeeds with probability $h = 0.8$ if agent $i$ exerts effort and with probability $l = 0.1$ if they do not. Every episode, each agent $i$ takes one of two actions, either exert effort or not. We sample the success of each task $t_i$ as a Bernoulli variable, with a success probability of either $h$ or $l$, depending on agent $i$'s action. The entire project is successful only if all tasks $\{t_i\}_{i=1}^n$ are successful. The rewards $r = (r_1, \ldots, r_n)$ are the parameters of our environment; an agent who exerts effort incurs a negative reward $-c$, and if the project is successful they also receive a reward $r_i$. Table 1 shows the possible reward schemes for these settings. We briefly discuss how these were computed, with full details in the appendix.

| Reward scheme | Rewards $r = (r_1, r_2, \ldots, r_5)$ |
|---|---|
| Nash scheme | $(35 + \epsilon, 35 + \epsilon, 35 + \epsilon, 35 + \epsilon, 35 + \epsilon)$ |
| Dominant Scheme | $(142, 857 + \epsilon, 142, 857 + \epsilon, 142, 857 + \epsilon, 142, 857 + \epsilon, 142, 857 + \epsilon)$ |
| Iterated dominance scheme | $(142, 857 + \epsilon, 17, 858 + \epsilon, 2, 233 + \epsilon, 280 + \epsilon, 35 + \epsilon)$ |

Table 1: Reward schemes in our joint project principal-agent environment.

Consider agent $i$ who is promised a reward $r_i$ and who knows that *exactly* $m$ of the other agents would exert effort (so the remaining $n - m - 1$ will not exert effort). If $i$ exerts effort, the project succeeds with probability $h^{m+1} \cdot l^{n-m-1}$, so their expected reward is $h^{m+1} \cdot l^{n-m-1} \cdot r_i - c$. If $i$ does not exert effort, the project succeeds with probability $h^m \cdot l^{n-m}$, and their expected reward is $h^m \cdot l^{n-m} \cdot r_i$. Agent $i$ would thus exert effort if: $h^{m+1} \cdot l^{n-m-1} \cdot r_i - c > h^m \cdot l^{n-m} \cdot r_i$, or equivalently if $r_i > c/(h^{m+1} \cdot l^{n-m-1} - h^m \cdot l^{n-m})$. Observe that the minimal reward to induce $i$ to exert effort decreases in $m$, and when $i$ assumes *no* other agents would exert effort ($m = 0$), the required reward $r_i$ is $r_i > c/(h \cdot l^{n-1} - l^n)$. Thus setting $r_i$ to $r_i = c/(h \cdot l^{n-1} - l^n) + \epsilon$ for all agents makes exerting effort a *dominant strategy* for all agents. In contrast, when $i$ assumes all other agents exert effort ($m = n - 1$), the required reward $r_i$ is $r_i > c/(h^n - h^{n-1} \cdot l)$, so setting $r_i = c/(h^n - h^{n-1} \cdot l) + \epsilon$ for all agents makes exerting effort a *Nash equilibrium*. Setting $r_i = c/(h^{m+1} \cdot l^{n-m-1} - h^m \cdot l^{n-m}) + \epsilon$ results in an *iterated dominance scheme*: the dominant strategy for the first agent is exerting effort; once not exerting effort has been eliminated as a strategy for player $i - 1$, player $i$ assumes players 1 to $i - 1$ would exert effort, and thus they also exert effort.

As Table 1 shows, even with only five agents, there are huge differences in the principal's expenditure when the project succeeds. We simulate the environment with all three reward schemes, setting $\epsilon = 160$ so the reward is just above the minimum threshold ($\epsilon = 160$ is negligible compared to the high reward $142, 857$), and use both REINFORCE learners, and Advantage Actor-Critic agents (Mnih et al., 2016) agents. Our results show that under the cheap Nash scheme MARL does not converge to exerting effort (rather, all agents end up *not* exerting effort). However, MARL does converge on exerting effort for the iterated dominance scheme, which is far cheaper than the dominant scheme.

Figure 1 shows the proportion of times where agents select the high effort action over training time under the Nash scheme, indicating that agents do not converge on the Nash equilibrium of all exerting effort. Figure 2 shows the results under the dominant strategy scheme, showing all agents converge on exerting effort. Figure 3 shows the results for the iterated dominance reward scheme, for different

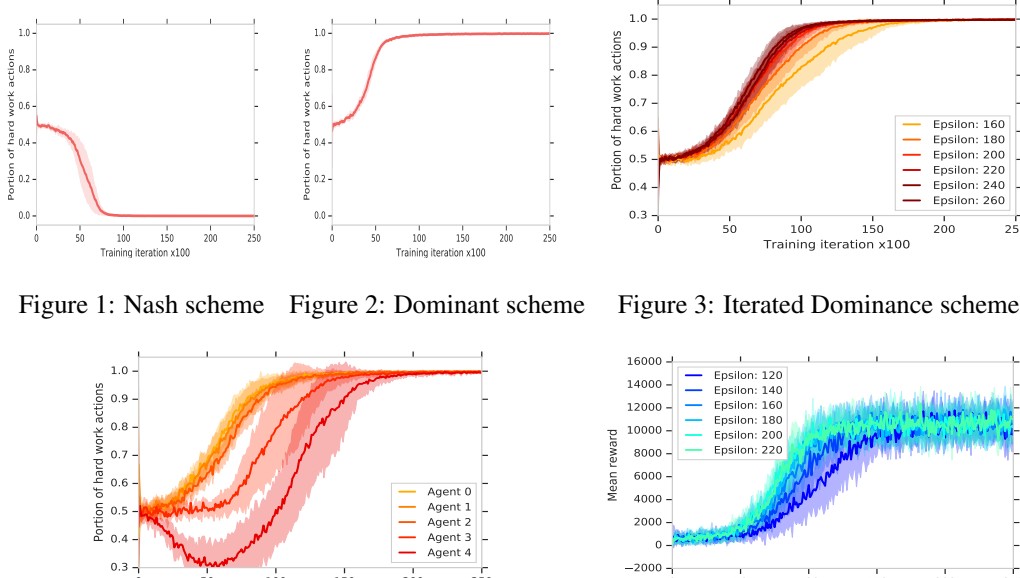

Figure 1: Nash scheme    Figure 2: Dominant scheme    Figure 3: Iterated Dominance scheme

Figure 4: Individual agent effort (Iterated Dominance)   Figure 5: Rewards (Iterated Dominance)

values of $\epsilon$. It shows that agents indeed learn to all exert effort, at a much lower cost to the principal than in the dominant strategy scheme (roughly 20% of the cost under the dominant strategy scheme).

Figure 4 shows the effort level of individual agents over training in the iterated dominance scheme (measured by the proportion of times where each agent selected the high effort action). It shows that first the highest reward agent learns to exert effort, then the next agent and so on. Interestingly, given the initial effort levels of the other agents, the last agent (with smallest promised reward) initially learns not to exert effort. Only after the other agents exert more effort, this agents learns to exert more effort. Figure 5 shows the mean agent reward over time in the iterated dominance scheme (similarly to Figure 3). It shows that as agents exert more effort, they improve their overall reward (reaching the reward under the scheme when all agents exert effort). The above figures are for REINFORCE agents, but our experiments with Advantage Actor Critic agents (Mnih et al., 2016) yield very similar results.

These results highlight the importance of the iterated dominance concept for multiagent systems comprised of independent reinforcement learners: such systems may not converge to the desired Nash equilibrium, but do converge to an iterated dominance solution. Thus, when designing mechanisms for multiple reinforcement learners, one should strive for an implementation that is based on the stronger iterated dominance solution, rather than on the less demanding Nash equilibrium.

We note that our theoretical results hold for REINFORCE only when agents have two actions, however, in the appendix we consider a simulation with three actions (effort levels), and we show that empirically agents do converge to the desired outcome in this case as well.

## 5 CONCLUSION

We have provided convergence results for MARL in iterated dominance solvable games, and discussed their implications to mechanism design for RL agents. Our results show that reward schemes based on iterated dominance are desirable, as MARL with reasonable learning methods is guaranteed to converge to such a solution, in contrast to schemes based on a Nash equilibrium. Several directions are open for future research. First, while we only proved convergence for specific RL algorithms or under some restrictions on the underlying game, we conjecture convergence occurs in wider settings. Could our results be extended to cover other RL algorithms or fewer restrictions on the game? In particular, can one prove convergence for REINFORCE with three or more actions? Second, we have focused on strict dominance — what can one say about weak iterated dominance? Finally, could we theoretically bound the required time to convergence to an iterated dominance solution?

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

## 6 APPENDIX: RL IN THE PRINCIPAL-AGENT JOINT PROJECT DOMAIN

Our empirical results regarding the joint project domain show how our theoretical results regarding MARL in iterated dominance solvable games can be used for mechanism design for reinforcement learners. We make the point that while many games are not iterated dominance solvable, in various domains we may *design* the game the agents play. We demonstrated how an iterated dominance design is preferable to a design based on the less demanding Nash equilibrium using the the principal-agent domain. A principal trying to cut costs (promised rewards) may settle on a *Nash reward scheme*, where it is a Nash equilibrium for all agents to exert effort. On the other hand, a principal who is extremely cautious may try and guarantee that all agents exert effort, by using a *dominant strategy scheme*, which promises extremely high rewards, sufficient to make it a *dominant* strategy for *every* agent to exert effort. Our empirical results show that when agents are "reasonable" reinforcement learners, neither extreme is recommended. For RL agents, the principal could promise an *iterated dominance* reward scheme, which on the one hand achieves convergence to the desired outcome of exerting effort, and on the other hand does so at a far lower cost than the dominant reward scheme.

The empirical results in the main paper consider a setting where agents have two effort levels (exerting high effort or exerting no effort) and where the joint project only succeeds when all tasks succeed. In this appendix we provide more detailed experimental results:

1. We provide a more detailed derivation of the reward schemes: the Nash scheme, the dominant strategy scheme, and the iterated dominance scheme.

2. We consider a richer model with **three effort levels**: no effort, medium effort and high effort (each resulting in a different task success probability). We show that REINFORCE agents converge to the desired outcome of exerting effort. Our theoretical results for REINFORCE only hold for the two action case, so this gives some indication that convergence occurs even in settings where it is not *guaranteed* by our theoretical results.

3. We discuss a more general **combinatorial joint project model**, where the entire project may succeed even when some tasks fail. In this case, we have a list of task subsets which result in the successful completion of the project. We provide simulation results for the case where the project succeeds if all tasks are successful, or if at most one task fails. Our results show that in this case as well, agents converge to the desired outcome of exerting effort under an iterated dominance reward scheme.

We begin by presenting the more general model of the joint project problem, and discuss how the three reward schemes are derived. Finally, we provide the additional empirical results discussed above.

The **joint project principal-agent domain** relates to a project comprised of a set of tasks $T = \{t_i\}_{i=1}^n$. In the main paper we considered the case where the project is successful only when all tasks are successful. In the more general model, the joint project is successful when certain subsets of tasks are successful. We consider a *technology function* $v : \mathbb{P}(T) \to \{0, 1\}$ which maps a subset of tasks to a Boolean value indicating whether completing them results in successfully completing the project. We generally assume the technology function is monotone so if $t_1 \subseteq t_2 \subseteq T$ and $v(t_1) = 1$ then $v(t_2) = 1$. We refer to a subset of tasks $T'$ such that $v(T') = 1$ as a set of tasks *fulfilling* the project. The set of all task subsets fulfilling the project is denoted as $T_{Win} = \{T_w \subseteq T | v(T_w) = 1\}$.

A set of agents $I = \{a_i\}_{i=1}^n$ are in charge of the tasks, with agent $a_i$ in charge of task $t_i$. Task $t_i$ has a base probability $l_i > 0$ of succeeding, and agent $i$ has two strategies $S_i = \{e, d\}$ where $e$ stands for exerting high effort, and $d$ stands for not exerting effort. If $a_i$ exerts effort, the probability of task $t_i$ succeeding rises from its base probability of $l_i$ to a higher level $h_i \leq 1$. The additional effort is costly, incurring negative reward $c_i < 0$ to the agent. We assume the tasks succeed or fail independently of one another.

For a subset of players $C \subset I$, we denote the tasks owned by these players $T(C) = \{t_i \in T | a_i \in C\}$. Suppose agents in $C$ exert effort, and agents in $I \setminus C$ do not exert effort. If $a_i \in C$ then $t_i$ succeeds with probability $h_i$, and if $a_i \notin C$, $t_i$ succeeds with probability $l_i$. Given $C \subseteq I$, we denote the probability task $t_i$ is successful as $p_i(C)$, where

$$p_i(C) = \begin{cases} h_i & \text{if } t_i \in C \\ l_i & \text{if } t_i \notin C \end{cases}$$

Consider a subset of tasks $T' \subset T$. When only the players in $C \subseteq I$ exert effort, the probability that *exactly* the tasks in $T'$ are successful is:

$$\Pr_C(T') = \prod_{t_i \in T'} p_i(C) \cdot \prod_{t_i \notin T'} (1 - p_i(C))$$

Similarly, the the probability that *any* subset of tasks fulfilling the project is achieved is $Pr_C(Win) = \sum_{T_w \in T_{win}} Pr_C(T_w)$.

To incentivize agents to exert effort, and maximize the probability of successfully completing the overall project, the principal offers the agents a reward vector $r = (r_1, \ldots, r_n)$, conditioned on the project's success. If the project is successful, agent $i$ receives a reward of $r_i$ from the principal (and receive nothing if the project fails).

The **joint project game** is a normal form game $G(r)$ obtained given the specific reward vector $r$ chosen by the principal. When only the agents in $C \subset I$ exert effort (i.e. any $a_i \notin C$ does not exert effort), $a_i$'s expected reward from the principle is:

$$e_i(C) = \sum_{T_w \in T_{win}} \Pr_C(T_w) \cdot r_i$$

Agent $a_i$ has a cost $c_i$ for exerting effort, and can choose between two strategies: exerting effort ($e$), or not exerting effort ($d$), where exerting effort increases the probability of the project succeeding and thus the expected reward from the principal, but at a the certain cost $c_i$. We denote by $S$ the set of possible strategy profiles $S = S_1 \times \ldots \times S_n$, where for any agent $i$ we have $S_i = \{e, d\}$. Given a strategy profile $s = (s_1, \ldots s_n) \in S$, we denote the set of agents who exert effort in $s$ by $C_s = \{a_i \in I | s_i = e\}$. The payoffs $u_i : S \to \mathbb{R}$ have a simple structure (note that setting $s_i = e$ increases the project success probability and thus the expected reward $e_i(C_s)$):

$$u_i(s) = \begin{cases} e_i(C_s) - c_i & \text{if } s_i = e \\ e_i(C_s) & \text{if } s_i = d \end{cases}$$

## 6.1 Appendix: A Detailed Description of the Joint Project Reward Schemes

We give a more detailed derivation of the reward schemes for the principal in the joint project problem: the Nash equilibrium scheme, the dominant strategy scheme, and the iterated dominance scheme. A *Nash scheme* is a reward scheme $r = (r_1, r_2, \ldots, r_n)$ where the strategy profile where all agents exert effort is a Nash equilibrium. Given the definition of a Nash equilibrium, this means that no agent wants to unilaterally deviate. In other words, the requirement is that for any agent $i$, *under the assumption* that all other agents are going to exert effort, agent $i$ would also rather exert effort. [1] Such Nash based schemes are widely studied in the mechanism design literature, with some work specifically discussing such schemes in joint project principal-agent settings (Babaioff et al., 2006).

In contrast, a *dominant strategy* scheme is a reward scheme $r = (r_1, r_2, \ldots, r_n)$ where the dominant strategy of every agent is exerting effort. In other words, the requirement here is that *no matter what the agent assumes others would do*, they would rather exert effort.

We first discuss the model used in the empirical section of the main paper, where the joint project requires the success of all $n$ tasks, each of which succeeds with a probability $h$ when the relevant agent exerts effort and probability $l$ when the relevant agents does not exert effort.

An agent's expected reward $u_i(s)$ depends on its chosen action $s_i$ as well as the actions of other agents $s_{-i}$. In our simple joint project model, all tasks have a success probability $h$ when the relevant agent exerts effort, and a success probability of $l$ if they do not exert effort. As these probabilities $l, h$ are the same for all tasks, it is sufficient to know *how many* agents exert effort in the profile $s$ to know the success probability of the project (we do not need to know exactly *which* agents exerted effort).

Consider agent $i$ who is promised a reward $r_i$ and who knows that exactly $m$ of the remaining agents are going to exert effort (and the remaining $n - m - 1$ will not exert effort).

---

[1] Note that there is no requirement for this to be the *only* Nash equilibrium.

If agent $i$ exerts effort, the probability of the project succeeding and the agent getting the reward $r_i$ is $h^{m+1} \cdot l^{n-m-1}$, and their expected reward is then $h^{m+1} \cdot l^{n-m-1} \cdot r_i - c$.

Similarly, if agent $i$ does not exert effort, the project succeeds with probability $h^m \cdot l^{n-m}$, and their expected reward is then $h^m \cdot l^{n-m} \cdot r_i$.

Agent $i$ would exert effort if their expected reward when exerting effort is high than the expected reward when not exerting effort:

$$h^{m+1} \cdot l^{n-m-1} \cdot r_i - c > h^m \cdot l^{n-m} \cdot r_i$$

Equivalently, the condition for exerting effort is $h^{m+1} \cdot l^{n-m-1} \cdot r - h^m \cdot l^{n-m} \cdot r > c$. We extract the threshold reward for exerting effort, as a function of $m$:

$$r_i(m) > c/(h^{m+1} \cdot l^{n-m-1} - h^m \cdot l^{n-m})$$

Observe that the threshold reward to induce $i$ to exert effort, $r_i(m)$ is monotonically decreasing in $m$, so the more agents $i$ assumes would exert effort, the less it needs to be rewarded to induce it to exert effort. Given the above function $r_i(m)$, the threshold reward to induce agent $i$ to exert effort when assuming all other agents would not exert effort ($m = 0$) is: $r_i > c/(h \cdot l^{n-1} - l^n)$. As $r_i(m)$ is decreasing in $m$, setting $r_i > c/(h \cdot l^{n-1} - l^n)$ is sufficient to induce $i$ to exert effort for any $m \in \{0, 1, \ldots, n-1\}$. In other words, by setting $r_i > c/(h \cdot l^{n-1} - l^n)$, we induce $i$ to exert effort for any strategy profile $s_{-i} \in S_{-i}$, making it the dominant strategy for $i$ to exert effort. Thus, setting $r_i > c/(h \cdot l^{n-1} - l^n)$ for any agent $i$ is a **dominant strategy scheme**.

In contrast, when $i$ assumes all other agents would exert effort ($m = n - 1$), the threshold reward $r_i$ given the above function $r_i(m)$ is:

$$r_i > c/(h^n - h^{n-1} \cdot l)$$

Setting the above reward $r_i > c/(h^n - h^{n-1} \cdot l)$ for all agents makes it a Nash equilibrium to exert effort; when agent $i$ assumes that all other agents exert effort, it assumes that $m = n - 1$, and given its own reward $r_i$ it would rather exert effort than not; thus, when all agent are exerting effort, no agent has an incentive to *unilaterally* deviate and stop exerting effort. Setting $r_i > c/(h^n - h^{n-1} \cdot l)$ is therefore a **Nash reward scheme**.

We now consider constructing a reward scheme based on iterated dominance. Given the discussion above, if $r_1 > c/(h^{n-1} - h^{n-1} \cdot l)$, agent 1 would exert effort if they assume all the remaining $m = n - 1$ do not exert effort. As the threshold reward $r_1(m)$ to induce agent 1 to exert effort when they assume exactly $m$ other agents would exert effort is diminishing in $m$, this means that when setting $r_1 > c/(h^{n-1} - h^{n-1} \cdot l)$ makes exerting effort a dominant strategy for agent 1. We can now turn to agent 2. First, observe that if $r_1 > c/(h^{n-1} - h^{n-1} \cdot l)$, agent 2 can eliminate not exerting effort as a strategy for agent 1. When agent 2 assumes agent 1 would exert effort, they rule out the value $m = 0$ (when at least one other agent exerts effort, the number $m$ of other agents exerting effort has to be at least $m \geq 1$). As agent 2 can assume at least one agent (agent 1) would exert effort, we can set $r_2 > c/(h^2 \cdot l^{n-2} - h \cdot l^{n-1})$, and under iterated elimination of dominated strategies, agent 2 would exert effort. Following the same argument for agents $3, 4, \ldots n$, we can construct an **iterated dominance reward scheme** by setting:

$$r_i > c/(h^{m+1} \cdot l^{n-m-1} - h^m \cdot l^{n-m})$$

The discussion above shows that the only action profile surviving iterated elimination of strictly dominated strategies being exerting effort. Note that this scheme means that each agent gets a different reward, despite the symmetry in task success probabilities.

As we note in the main text, the total amount spent by the principal depends on the reward scheme used. Our simulations were based on an environment with five agents, an effort exertion cost of $c = 10$, and task success probabilities $l = 0.1$ and $h = 0.8$. As discussed above, the Nash scheme requires setting $r_i > c/(h^n - h^{n-1} \cdot l)$ for all agents, so under these parameters $r_i = 35 + \epsilon$ is a Nash scheme, which is a very low total payment to the principal. However, our empirical analysis revealed that MARL does not converge to the desired outcome of exerting effort, but rather on *not* exerting effort.

In contrast, setting $r_i > c/(h \cdot l^{n-1} - l^n)$ for all agents is a dominant strategy scheme, so under the above parameters, setting $r_i = 142, 857 + \epsilon$ for all agents results in a exerting effort being a dominant strategy. This dominant strategy implementation is much more costly to the principal, but agents are guaranteed to converge to exert effort (as under this payment scheme, it is a dominant strategy to do so).

An iterated dominance scheme is far less costly than the dominant strategy scheme. As discussed above, in an iterated dominance scheme, agents are promised different rewards, as $r_i > c/(h^{m+1} \cdot l^{n-m-1} - h^m \cdot l^{n-m})$. For the settings above, the scheme offers rewards of $r = (r_1, r_2, \ldots, r_5) = (142, 857 + \epsilon, 17858 + \epsilon, 2233 + \epsilon, 280 + \epsilon, 35 + \epsilon)$. While this is far cheaper than the dominant strategy scheme, our theoretical results show that independent MARL would converge to all agents exerting effort.

The main text of the paper shows the empirical results for the above parameters, showing that MARL converges to the desired outcome of all agents exerting effort under the iterated dominance scheme, as well as the dominant scheme, but that agents end up exerting no effort under the Nash scheme.

## 6.2 THREE EFFORT LEVELS

So far we considered the case where every agent had two strategies: exert effort, or not exert effort. Exerting no effort had zero cost, while exerting effort had a cost $c = 10$. we now consider the case where there is an **intermediate effort level**, whose cost to the agent is $c_d$ where $0 < c_d < c$ (for instance $c_d = 6$), and which results in an intermediate probability $d$ for success in the relevant task, so $l = 0.1 < d < h = 0.8$ (for instance, $d = 0.4$).

In particular, we note that our results int the main paper on MARL converging to the iterated dominance solution under REINFORCE relate to games with at most two actions per agent, and thus do not hold for this case: for domains with more than two actions, we only proved convergence is guaranteed under the the importnace weighted version of MC policy improvement (IW-MCPI). We now analyze this setting empirically.

As before, to make exerting high effort a Nash equilibrium, we require the expected reward when exerting high effort to be higher than not exerting effort (assuming other agents are all exerting high effort): $h^n r - c > h^{n-1} lr$ or equivalently $r > \frac{c}{h^n - h^{n-1} l}$. However, we also have the additional condition that exerting high effort is preferable to exerting medium effort: $h^n r - c > h^{n-1} dr - c_d$ or equivalently:

$$r > \frac{c - c_d}{h^n - h^{n-1} d}$$

Similarly, when assuming exactly $m$ of the other agents would exert high effort, an agent would rather exert high effort than not exerting any effort when $r_m > c/(h^{m+1} \cdot l^{n-m-1} - h^m \cdot l^{n-m})$ (we call this condition I). Using similar arguments, when assuming exactly $m$ of the other agents would exert high effort, an agent would rather exert high effort than exerting medium effort when $r_m > \frac{c - c_d}{h^m l^{n-m} \cdot (h-d)}$ (we refer to this as condition II). Both condition I and condition II place a requirement on the reward for the $m$'th agent in an iterated dominance solution. Whether condition I is more demanding than condition II or vice versa depends on the domain parameters $h, d, l, c, c_d$. We note that for the setting discussed above where $c_d = 6$ and $d = 0.4$ (and the remaining parameters of $l = 0.1, h = 0.8, c = 10$ as before), the more demanding condition is condition I, so the earlier iterated dominance reward scheme of $r = (r_1, r_2, \ldots, r_5) = (142, 857 + \epsilon, 17858 + \epsilon, 2233 + \epsilon, 280 + \epsilon, 35 + \epsilon)$ also applies in this settings, making the strategy profile where all agents exert high effort the iterated dominance solution. Also, similarly to before, setting $r = 35 + \epsilon$ for all agents (for $\epsilon < 240$) makes exerting high effort a Nash equilibrium, and setting $r > 142, 857 + \epsilon$ for all agents makes exerting high effort a dominant strategy for all agents.

Figure 6, 7 and 8 show the proportion of time that agents exert high effort over training time in the 3 effort level case (similar to the figures in the main paper regarding the two action case). The results here are very similar to the two action domain: agents converge on exerting effort in both the dominant strategy scheme and iterated dominance scheme, but not in the Nash scheme.

We again emphasize that, as opposed to the two action case, our theoretical results do not *guarantee* convergence in this setting for REINFORCE. However, given these encouraging empirical results, we

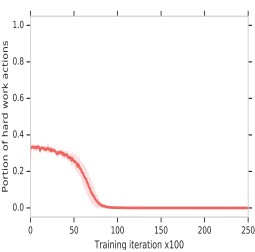 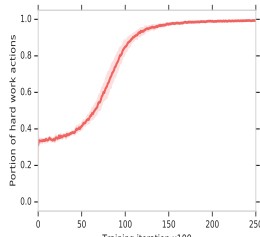 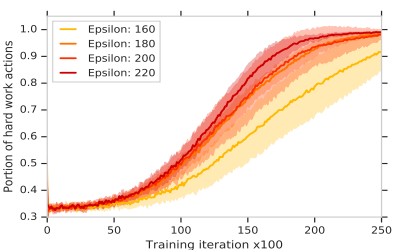

Figure 6: Nash scheme   Figure 7: Dominant scheme   Figure 8: Iterated Dominance scheme

conjecture that MARL converges to iterated dominance solutions under less restrictive conditions than those we used to prove our theoretical results (i.e. for less demanding algorithms, or for REINFORCE or other known simple RL algorithms in less restricted classes of games).

### 6.2.1 JOINT PROJECTS WITH ALLOWED TASK FAILURES

So far we have focused on the restricted case where the entire joint project is successful only if *all* the tasks are successful. We now consider the case where there is some redundancy between tasks, so the project may succeed even if some tasks fail. Specifically, we examine the setting where the project succeeds if at most one task fails. In the case of $n = 5$ tasks, this means we model the project as successful if at least $n - 1 = 4$ tasks succeed. We still use the same parameters as before: a cost $c = 10$ for exerting effort, and a success probability of $l = 0.1$ for a task when the relevant agent exerts no effort and $h = 0.8$ when the relevant agent does exert effort.

We note that the computation of the Nash, dominant strategy and iterated dominance schemes become slightly more elaborate in this case. We briefly discuss computing reward schemes for this case. An agent takes into account the probability of the overall project succeeding when they exert effort and when they do not. When the agent assumes exactly $m$ other agents would exert effort, the probability of the project succeeding assuming they *do* exert effort is:

$$p(e, m) = (1-h) \cdot h^m \cdot l^{n-m-1} + h \cdot h^m \cdot l^{n-m-1} + h \cdot m \cdot (1-h) \cdot h^{m-1} \cdot l^{n-m-1} + h \cdot (n-m-1) \cdot (1-l) \cdot h^m \cdot l^{n-m-2}$$

Similarly, if the agent does *not* exert effort, and exactly $m$ of the other agents exert effort, the probability of the project succeeding is:

$$p(s, m) = (1-l) \cdot h^m \cdot l^{n-m-1} + l \cdot h^m \cdot l^{n-m-1} + l \cdot m \cdot (1-h) \cdot h^{m-1} \cdot l^{n-m-1} + l \cdot (n-m-1) \cdot (1-l) \cdot h^m \cdot l^{n-m-2}$$

The above calculations are based on the following possibilities leading to the overall project succeeding: the event where the agent's task failed but all the other tasks succeeded, the event where all the tasks (the agent's and the other agents' tasks) succeeded, the case where the agent's task succeeded and the only failure was in a task where the other relevant agent was exerting high effort, and the case where the agent's task succeeded and the only failure was in a task where the other relevant agent was exerting no effort.

Similarly to the reward scheme computation discussed in the main text, we can compute three reward schemes for this setting. For a dominant strategy scheme we set $r_i = c/(p(e,0) - p(s,0)) + \epsilon$ for all agents, for a Nash scheme we set $r_i = c/(p(e, n-1) - p(s, n-1)) + \epsilon$ for all agents, and for an iterated dominance scheme, we set the reward of agent $i$ to be $r_i = c/(p(e, i-1) - p(s, i-1)) + \epsilon$. Under our parameters, the dominant strategy scheme is thus $r_i = 3968.25 + \epsilon$, the Nash scheme is $r_i = 35 + \epsilon$, and the iterated dominance scheme is $r = (r_1, r_2, \ldots, r_5) = (3968.25 + \epsilon, 655.30 + \epsilon, 120.65 + \epsilon, 28.62 + \epsilon, 35 + \epsilon)$.

We now present figures of agent effort (proportion of selecting the high effort action) over training times for this setting, under the three reward schemes. As the overall scale of rewards is much lower than in the setting where all tasks must succeed, we used smaller values of $\epsilon$.

Figures 9, 10 and 11 are similar to those for the previous settings: MARL under the Nash scheme fails to converge to exerting effort, but does converge to the desired outcome under both the expensive dominant strategy scheme and the far cheaper iterated dominance scheme. These results show some robustness of our results to the technology function used for the joint project principal agent domains.

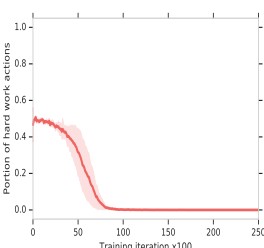 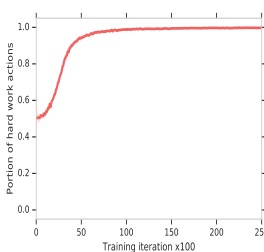 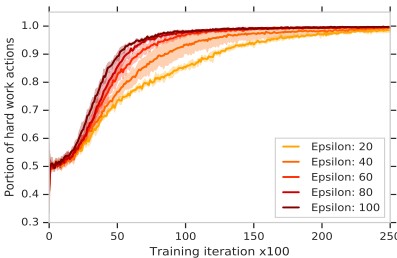

Figure 9: Nash scheme    Figure 10: Dominant scheme  Figure 11: Iterated Dominance scheme

### 6.3 MARL IN MARKOV GAMES (ENVIRONMENTS WITH MULTIPLE TIMESTEPS

Our analysis in the main paper has focused on normal-form games. However, many environments relate to temporally extended interaction between agents. A key model for such repeated multi-agent interaction across multiple timesteps is that of **Markov games** (Shapley, 1953; Littman, 1994). In Markov games, in each state agents take actions (possibly based only on partial observations of the true world state), and each agent obtaining an individual reward. One prominent method for applying multi-agent learning in such settings is that of independent MARL, where agents each learn a behavior policy through their individual experiences interacting with one another in the environment.

Formally, one may consider an $n$-player Markov game $\mathcal{M}$ (Shapley, 1953; Littman, 1994) defined on a finite state set $\mathcal{S}$. An observation function $O : \mathcal{S} \times \{1, \ldots, N\} \rightarrow \mathbb{R}^d$ gives each agent's $d$-dimensional restricted view of the true state space. On any state, each agent applies an action from $\mathcal{A}^1, \ldots, \mathcal{A}^N$ (one per agent). Given the joint action $a^1, \ldots, a^N \in \mathcal{A}^1, \ldots, \mathcal{A}^N$ the state changes, following a transition function $\mathcal{T} : \mathcal{S} \times \mathcal{A}^1 \times \cdots \times \mathcal{A}^N \rightarrow \Delta(\mathcal{S})$ (this allows for a stochastic transition, and we denote the set of probability distributions over $\mathcal{S}$ as $\Delta(\mathcal{S})$). By $\mathcal{O}^i = \{o^i \mid s \in \mathcal{S}, o^i = O(s, i)\}$ we denote the observation space of agent $i$. Each agent $i$ gets an individual reward $r^i : \mathcal{S} \times \mathcal{A}^1 \times \cdots \times \mathcal{A}^N \rightarrow \mathbb{R}$.

Each agent has its own experience in the environment, and independently learns a policy $\pi^i : \mathcal{O}^i \rightarrow \Delta(\mathcal{A}^i)$ (denoted $\pi(a^i|o^i)$) given its own observation $o^i = O(s, i)$ and reward $r^i(s, a^1, \ldots, a^N)$). We use the notation $\vec{a} = (a^1, \ldots, a^N)$, $\vec{o} = (o^1, \ldots, o^N)$ and $\vec{\pi}(.|\vec{o}) = (\pi^1(.|o^1), \ldots, \pi^N(.|o^N))$. Every agent attempts to maximize its long term $\gamma$-discounted utility:

$$V_{\vec{\pi}}^i(s_0) = \mathbb{E}\left[\sum_{t=0}^{\infty} \gamma^t r^i(s_t, \vec{a}_t)|\vec{a}_t \sim \vec{\pi}_t, s_{t+1} \sim \mathcal{T}(s_t, \vec{a}_t)\right] \qquad (4)$$

In a Markov game, we denote the set of all possible (deterministic) policies that agent $i$ can use as $\Pi_i$, relating to the set of all possible functions $\pi^i : \mathcal{O}^i \rightarrow \Delta(\mathcal{A}^i)$.

Our definition for a strategy $s_x$ strictly dominating strategy $s_y$ relates to the strategy $s_x$ yielding agent $i$ a higher utility than $s_y$ no matter what the other agents do. This definition relates to normal form game, rather than the more general Markov game setting. However, this definition, as well as the iterated dominance solution concept, can easily be adapted to apply to Markov games. Given a Markov game setting we identify the set $S_i$ of strategies available to each agent $i$ as the set $\Pi_i$ of all policies agent $i$ has in the Markov game, and identify the payoff for agent $i$ in the game under the policies $\vec{\pi} = (\pi_1, pi_2, \ldots, \pi_n)$ with the expected long term $\gamma$ discounted utility, so $u_i(\vec{\pi}) = V_{\vec{\pi}}^i(s_0)$.

Then, under the standard definition of strategy domination, we say policy $\pi_x^i \in \Pi_i$ strictly dominates policy $\pi_y^i \in \Pi_i$ if for any joint policy the other agents may use $\pi_{-i}$, the policy $\pi_x^i$ achieves a higher utility than $\pi_y^i$, so that for any $\pi_{-i} \in \Pi_{-i}$ we have $V_{(\pi_x^i, \pi_{-i})}^i(s_0) > V_{(\pi_y^i, \pi_{-i})}^i(s_0)$. Hence the definition of iterated dominance can be used in the setting of Markov games as well. [2]

---

[2]Note that the above definition considers deterministic policies, yielding a finite set of strategies. While many RL algorithms use stochastic policies, when agent $i$ is responding to fixed policies of other agents, the optimal policy is a deterministic one.

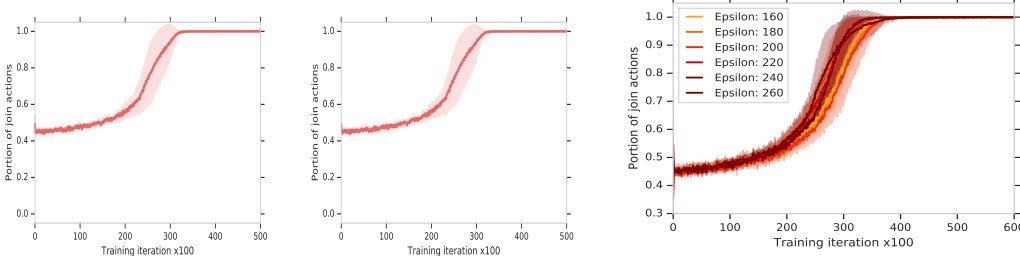

Figure 12: Nash schemeFigure 13: Dominant scheme Figure 14: Iterated Dominance scheme
Opt-in Proportion         Opt-in proportion         Opt-in proportion

To demonstrate the how our work applies to a Markov game setting, we consider an environment similar to the principal-agent joint project setting discussed in the main text, but with multiple time-steps.

Each episode in our environment has two time-steps. In the first step, each agent has to decide whether they want to participate in the project, and has two actions: *opt-in* and *opt-out*. If even one agent opts-out, all agents incur a penalty $p$ (negative reward). If all agents opt-in, we proceed to the second time-step, in which agents make decisions about their effort level, exactly as given in the original game discussed in Section 4; Agents can either exert effort or not, with a cost $c = 10$ for exerting effort, with a probability of $h = 0.8$ for a task succeeding under high effort and a probability $l = 0.1$ for it succeeding under no effort (and with the project only succeeding when all tasks succeed). We examine the same reward schemes given in Table 1 (Nash scheme, Dominant Scheme and Iterated Dominance Scheme).

We note that by setting the opt-out penalty to $p = 15$ (i.e. opting out gives a reward of $-15$), any policy which opts out in the first step is dominated by any policy that opts-in during the first step — the worst possible outcome for an agent which opted-in is exerting effort and having the project fail, and even that givens a reward of $-10$, which is better than opting out. Hence, in this case, under the iterated dominance payment scheme of Table 1, we get a dominance solvable game where all agents opt-in and exert high effort. In contrast, under the Nash scheme of Table 1, one Nash equilibrium is having all agents opt-in but exerting no effort.

We now provide the simulation results for this setting, similarly to Figure 1 to Figure 5.

Figure 12, Figure 13 and Figure 14 show the proportion of agents who opt-in during the first timestep under the Nash, Dominant and Iterated Dominance rewrd schemes, respectively. The figures show that under all reward schemes, agents quickly learn to opt-in during the first timestep. This is unsurprising, due to the high opt-out penalty.

Figure 15, Figure 16 and Figure 17 show the proportion of agents who choose to exert effort during the second timestep under the Nash, Dominant and Iterated Dominance reward schemes, respectively. The figures show results similar to those obtained for the single timestep environment. Under the Nash scheme, all agents quickly learn not to exert effort. In contrast, under both the Dominant and Iterated Dominance schemes, all agents learn to exert effort during the second timestep (though the principal's payments are vastly different). In this case, agents learn to avoid the dominated policies starting with opting out, then learn to exert effort once they have opted-in.

For completeness, Figure 18 shows the proportion of high effort actions during the second timestep, and Figure 19 shows the the mean agent reward over training. Both figures show similar behavior to that in the single timestep environment.

The figures and discussion above show how our results relate to Markov games with multiple timesteps. In settings where one can identify a domination sequence over *policies* describing agent behavior in temporally extended environments, agents would likely follow the elimination sequence over policies, converging to an iterated dominance solution, if one exists.

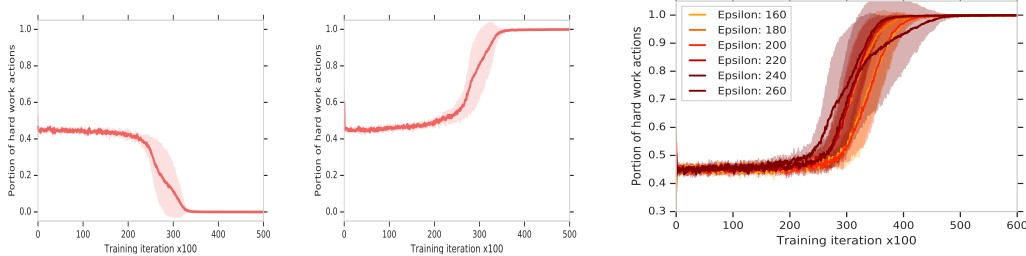

Figure 15: Nash scheme High effort proportion     Figure 16: Dominant scheme High effort proportion     Figure 17: Iterated Dominance scheme High effort proportion

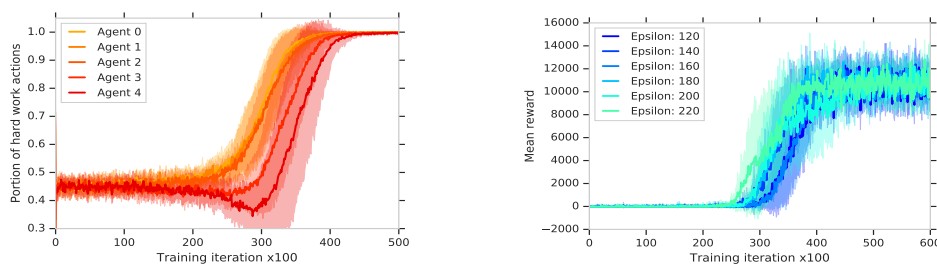

Figure 18: Individual agent effort (Iterated Dominance)     Figure 19: Rewards (Iterated Dominance)

## 7 CONVERGENCE RATES FOR IW-MCPI

Theorem 3.4 provides an asymptotic result showing that in the limit of infinite data IW-MCPI almost surely converges to a iterated elimination solution. The speed of convergence is game-dependent. The important factor is the degree to which dominated actions are suboptimal, which is captured by the $g$ in the proof of Theorem 3.4.

Suppose now that all dominated actions are at least $\epsilon$-suboptimal. To derive a rate of convergence we need to make each of the arguments in (2) and (3) finite-time. Suppose that $\epsilon_t = t^{-p}$ for $p \in (0, 1)$. Then by the proof of Theorem 3.3, the estimation error about the rewards is after $t$ rounds is

$$\tilde{O}(\sqrt{t^{1+p}}) .$$

Using the fact that $g \geq \epsilon$ by assumption it follows from (2) that $\tau_2 - \tau_1 = O(\epsilon^{-2/(1-p)})$. Since the proof is iterated over $\sum_{i=1}^{n} |S_i|$ actions, the number of rounds before all IW-MCPI strategies play an iterated elimination solution with high probability is

$$O \left( \sum_{i=1}^{n} \frac{|S_i|}{\epsilon^{2/(1-p)}} \right) .$$

By high probability here, we mean with probability that is no more than a constant times the amount of forced exploration $\epsilon_t$, which by assumption decays with $O(t^{-p})$. For example, when $p = 1/2$, the joint strategy of the algorithms will be an iterated elimination solution with probability at least $1 - O(t^{-1/2})$ once

$$t = \Omega \left( \sum_{i=1}^{n} \frac{|S_i|}{\epsilon^4} \right) .$$

## 8 CONVERGENCE ISSUES FOR REINFORCE WITH MORE THAN TWO ACTIONS

Here we argue that convergence of REINFORCE is not obvious when the number of actions is larger than two.

The key observation is that the gradient update does not always guarantee that scores of dominated actions decrease relative to non-dominated actions. To see this, consider the case where the first player has three actions $S_1 = \{1, 2, 3\}$. Then the expected REINFORCE update when the opponents are playing strategy $b$ is

$$x_{t+1,a} = x_{t,a} + \alpha(\nabla_x J^b)_a = x_{t,a} + \alpha p_{x_t}(a)(r_a^b - p_{x_t}^\top r^b). \tag{5}$$

Suppose that $a = 1$ is dominated. Our goal is to prove there exists an $a'$ such that almost surely,

$$\lim_{t \to \infty} x_{t,a'} - x_{t,1} = \infty.$$

Assume for a moment that there are just two actions. Then by the definition of dominance $r_1^b \leq r_2^b$ and one can easily see from the gradient calculation in (5) that $x_{t+1,1}$ is decreasing in expectation and $x_{t+1,2}$ is increasing, which leads to the desired behaviour. Unfortunately this is no longer true when there are more than two actions. It is still true that $r_1^b - p_{x_t}^\top r^b$ is negative, but the gradient can be negative for other actions as well. Furthermore, the scaling by $p_{x_t}(a)$ means that once $p_{x_t}(1)$ is small, if action $a = 2$ is also suboptimal, then

$$p_{x_t}(2)(r_2^b - p_{x_t}^\top r^b) \ll p_{x_t}(1)(r_1^b - p_{x_t}^\top r^b)$$

is possible and proving a separation between $x_{t,1}$ and $x_{t,a}$ for some $a > 1$ is apparently non-trivial.

As an example of this, consider the case of player 1 having 3 actions, where the reward from these actions depends on the action choice of player 2, with the rewards being $r^{b_1} = (0, 0.5, 1)$ (respectively for the player 1's actions) when player 2 takes the first action or being $r^{b_2} = (0, 0.5, 0.1)$ when player 2 takes the other action. In other words, in this case the first action is always dominated for the player 1, but depending on player 2's action, the optimal action for player 1 may be either its second or third action.

We show that performing the REINFORCE update for player 1 always reduces the logit of the first action. However, for some action logits, the update on $r^{b_1}$ increases the logit of action 3 and decreases that of action 2, where *the decrease in the logit of action 2 is larger than the decrease in the logit of the dominated action 1*. Similarly, the update on $r^{b_2}$ increases the logit of action 2 and decreases that of action 3, where *the decrease in the logit of action 3 is larger than the decrease in the logit of the dominated action 1*.

More formally, consider the case of action logits of $x = (0, 1, 5)$, and $r^{b_1} = (0, 0.5, 1)$. Then

$$\nabla_x J^{b_1} \approx (-0.0065, -0.0087, 0.0151),$$

which means the scores of actions 1 and 2 are getting closer.

Now consider the case where the other player takes a different action, so we have $r^{b_2} = (0, 0.5, 0.1)$, so

$$\nabla_x J^{b_2} \approx (-0.0007, 0.00703, -0.00631),$$

so that now the scores of actions 1 and 3 are getting closer.

Although we could not find a *non*-convergence results, note that by oscillating between the two updates, one can keep all the action logits quite close together (and in particular, the probability of taking the dominate action is *not* monotinically decreasing).

## 9 COMPARISON OF OUR RESULTS AND EXISTING RESULTS ON THE CONVERGENCE OF OTHER ALGORITHMS TO THE ITERATED DOMINANCE SOLUTION

Our theoretical results study the convergence of REINFORCE and a version of importance weighted Monte-Carlo Policy Improvement to the iterated dominance solution in strict dominance solvable games. Iterated dominance is a key solution concept in game theory, and some earlier work has considered the convergence of other algorithms to the iterated dominance solution.

Earlier work has also shown a polynomial algorithm for computing the strict iterated dominance solution in dominance solvable games, but has shown that doing so for weak iterated-dominance is an NP-hard problem (Conitzer & Sandholm, 2005).

One algorithm that has been studied in relation to dominance solvable games is Fictitious Play (Brown, 1951). Fictitious Play is an iterative process designed to compute Nash equilibria, in which each player assumes the opponents are playing stationary mixed strategies. In every step, each player selects the best response to the historical empirical frequency of strategies played by their opponents. Though designed to compute a Nash equilibrium, Fictitious Play only converges to a Nash equilibrium in restricted classes of games, most notably two-player zero-sum games with a finite number of strategies (Robinson, 1951). Fictitious play has been shown to converge to a strict iterated dominance solution in dominance solvable games (Nachbar, 1990) (see various evolutionary game theory textbooks for a detailed discussion (Fudenberg et al., 1998; Weibull, 1997)).

Another evolutionary game theory dynamics that has been studied in relation to iterated dominance is the Replicator Dynamics (Taylor & Jonker, 1978; Schuster & Sigmund, 1983; Weibull, 1997), a model of the evolution of strategies in a population of agents participating in a game, captured by a differential equation called the replicator equation. In this model the relative frequencies of a strategy in a population evolve over time. In each period strategies are randomly matched against other players from the population based on the current frequencies, and the expected payoff (fitness) a strategy achieves determines its rate of reproduction (yielding the frequencies of strategies in the next step). The replicator dynamics also converges to a strict iterated dominance solution in dominance solvable games (Fudenberg et al., 1998; Bowling, 2000).

The above results on Fictitious Play and the Replicator Dynamics indicate that some adaptive procedures can identify a strict iterated dominance solution in dominance solvable games. However, convergence results on one type of algorithm may not apply to other types of algorithms. In other words, though different RL algorithms may sometimes converge to the same outcome (if they converge at all), this is certainly not an automatic guarantee. Hence, our results are not subsumed by earlier work. Further, there are inherent differences between Fictitious Play and the Replicator Dynamics and the RL algorithms we study, which may lead to different convergence behavior.

Our motivation is not to compute the iterated dominance solution, as there are already good algorithms for doing so (Conitzer & Sandholm, 2005). Rather, we want to study how commonly used RL algorithms behave in dominance solvable games. We focused on policy gradient methods as these lie at the heart of many popular agents, and on policy iteration as it is among the simplest and most basic methods.

As opposed to policy gradient methods like REINFORCE, Fictitious Play does not rely on computing gradients, but rather on repeatedly finding the best response to the empirical distribution of actions taken by the opponents so far. Similarly, the Replicator Dynamics remains a different dynamical system from policy gradient methods; The Replicator Dynamics is a *regret minimizing* approach which is known to have significant differences with policy gradient methods (Omidshafiei et al., 2019; Mertikopoulos et al., 2018). While we conjecture that other RL methods may also converge to an iterated dominance solution, there may not be an obvious way to extend such convergence results from one RL algorithm to another, making it an interesting open problem to identify RL algorithms that provably converge to iterated dominance solutions.

