# OpenReview forum: "Multiagent Reinforcement Learning in Games with an Iterated Dominance Solution"
_ICLR.cc/2020/Conference — Reject_

### Official Review · AnonReviewer1 · 2019-10-20
**Official Blind Review #1**

**Rating:** 1

**Review:**

This work studies learning under independent MARL, and shows theoretically and experimentally that two independent MARL algorithms converge for games that can be solved by iterated dominance.

This work is clear and well-written, but I do not understand what the contribution of this work is to the literature. The fact that standard MARL learning rules (e.g. independent Q learning) converge in games with iterated dominance solutions is a very well-known result in Learning in Games (see [1], [2]). The authors examined slightly different learning rules (REINFORCE and MCPI), but I would expect that almost any reasonable learning rule would converge in iterated-dominance-solvable games; if anything, it would be surprising if this were *not* the case. The applications of the convergence result result to "noisy effort" games is pretty standard and the results expected based on the theory.

Question to the authors:
- How does this work differ from the known results about convergence of naive learners in iterated-dominance-solvable games?

[1] Michael Bowling, "Convergence Problems of General-Sum Multiagent Reinforcement Learning", Sec. 5.2
[2] Fudenberg & Levine, 1999

**Experience Assessment:**

I have published one or two papers in this area.

**Review Assessment: Checking Correctness Of Derivations And Theory:**

I did not assess the derivations or theory.

**Review Assessment: Checking Correctness Of Experiments:**

I assessed the sensibility of the experiments.

**Review Assessment: Thoroughness In Paper Reading:**

I made a quick assessment of this paper.

---

> ### Author Response · Authors · 2019-11-14
> **The work you mention relates to Fictitious Play and Replicator Dynamics which are very different from the RL algorithms we consider. Our results are not subsumed by existing work.**
>
> Thank you for the comments, and for pointing out this related work.
>
> Our paper deals with Monte-Carlo Policy Improvement and REINFORCE (a policy gradient method). In general, convergence results on one type of algorithm may not apply to other types of algorithms. In other words, though different RL algorithms may sometimes converge to the same outcome (if they converge at all), this is certainly not an automatic guarantee.
>
> The results in Fudenberg and Levine [1] relate to Fictitious Play (FP) and the Replicator Dynamics (RD); as they mention, their results are actually due to Nachbar 1990, see Evolutionary selection dynamics in games: Convergence and limit properties. The work by Bowling [2] mentions Q learning, but merely says that in some cases multi-agent Q learning can find Nash equilibria, such as in fully collaborative games where all agents had identical rewards. It then points out that *other* naive learning methods (i.e. not Q-learning) converge to iterated dominance solutions, and point to the publication by Fudenberg and Levine. In short, these existing results apply to Fictitious Play and Replicator Dynamics, rather than the algorithms we considered, which are different (and again, all of these are different from Q learning), so our results are not covered by this existing work.
>
> Further, we note that the algorithms covered by this existing work (FP and RD) originate from evolutionary game theory, and were designed to compute equilibria. Thus, they are *fundamentally different* from the RL algorithms we consider. There exist good algorithms for computing iterated dominance solutions (see, e.g. Conitzer and Sandholm, Complexity of iterated dominance, 2005), but our motivation is not to compute the iterated dominance solution, but rather to study how commonly used RL algorithms behave in dominance solvable games. We focused on policy gradient methods as these lie at the heart of many popular agents, and on policy iteration as it is among the simplest and most basic methods.
>
> FP does not rely on computing gradients, but rather on repeatedly finding the best response to the empirical distribution of actions taken by the opponents so far. Similarly, RD remains a different dynamical system from policy gradient methods; RD is a *regret minimizing* approach which is known to have significant differences with policy gradient methods (see, e.g. Neural Replicator Dynamics, Omidshafiei et al. and Cycles in Adversarial Regularized Learning, Mertikopoulos, Papadimitriou and Piliouras). We are not aware of any results on the RL methods we study in this paper.
>
> In short, the results you mention deal with very different algorithms from those we consider (which are the ones relating to frequently used RL agents). We thus believe our results are novel and valuable to the RL community.
>
> Based on your comments, we are adding a section on “Comparison of Our Results and Existing Results on the Convergence of Other Algorithms to the Iterated Dominance Solution”, discussing how our results differ from the papers you discussed (and other work).
>
> Finally, you pointed out that you expect any “reasonable” learning rule would converge on this iterated dominance solution. To emphasize why the analysis is tricky, we have added an appendix discussing convergence issues with REINFORCE, which we briefly discuss here (see the new appendix “Convergence Issues for REINFORCE with More Than Two Actions” for full details).
>
> Consider the case of player 1 having 3 actions, where the reward from these actions depends on the action choice of player 2, with the rewards being r^b1 = (0, 0.5, 1) (respectively for the player 1’s actions) when player 2 takes the first action or being r^b2 = (0, 0.5, 0.1) when player 2 takes the other action. In other words, in this case the first action is always dominated for the player 1, but depending on player 2’s action, the optimal action for player 1 may be either its second or third action.
>
> Performing the REINFORCE update for player 1 always reduces the logit of the first action. However, for some action logits, the update on r^a increases the logit of action 3 and decreases that of action 2, where *the decrease in the logit of action 2 is larger than the decrease in the logit of the dominated action 1*. Similarly, the update on r^b increases the logit of action 2 and decreases that of action 3, where *the decrease in the logit of action 3 is larger than the decrease in the logit of the dominated action 1*.
>
> When player 2 constantly switches between the two actions, we oscillate between the first and second updates. This shows that the REINFORCE update does *not* guarantee that the logits of the dominated action decreases relative to that of the non-dominated actions, highlighting why the analysis is tricky.
>
> We hope the discussion above addresses your concerns regarding the novelty and technical contribution of the paper.

---

> > ### Comment · AnonReviewer1 · 2019-11-14
> > **Response**
> >
> > Hi, thanks for the detailed response. I still believe that the class of learning rules known to converge to Nash in iterated dominance (supermodular) games is known to be quite large. Lets look at what I believe is the original paper on this, [1] reading from (6A) to Theorem 8 and it's corollaries, which prove convergence to Nash in supermodular games for a very wide class of learning rules:
> >
> > "It requires only that, for any date T, there is a later date after which each player selects either a strategy that is "justifiable" in terms of the competitors' play since T ... Here, "justify"is used in a very weak sense. A strategy choice is justified if there is no other strategy that would have done better against every combination of strategies [from] the competitors' recent past play."
> >
> > Converting the terminology here to RL terminology, I believe it says:
> >
> > Any learning rule that eventually chooses only among actions that are the BR to *some* action played by the agents in the last T turns (for some T), will converge to Nash.
> >
> > Stating the converse of this:
> >
> > As long as a learning rule eventually stops playing actions that are the BR to *no* actions played in the last T turns (for some T), it will converge to Nash.
> >
> > I think all reasonable RL learning rules (all reasonable learning rules?) will have this property. Your example illustrates this: action 1 is the best response to *no* action played by B and therefore the learning rule will stop playing it.
> >
> > [1] http://www.parisschoolofeconomics.eu/docs/guesnerie-roger/milgromroberts90.pdf

---

> > > ### Author Response · Authors · 2019-11-14
> > > **This additional paper relates only to supermodular games, not any arbitrary game. Our results are for general games.**
> > >
> > > Thanks for reading our response and further discussion!
> > >
> > > The paper you cite focuses on supermodular games, which is a restricted class of games.
> > >
> > > As the authors themselves note (in the beginning of section 3, page 1268):
> > >
> > > "Recently, Fudenberg and Kreps (1988) have investigated limiting behavior in a class of learning models for general extensive form games. ...
> > > They conclude that learning may, even in the long-run, yield a larger set of strategies than is identified by Nash equilibrium.
> > > Shapley and Fudenberg-Kreps establish the rather negative conclusion that Nash equilibrium play is not the only possible outcome of learning in general  games. For *supermodular* games, however, sharper and more positive results are possible..."
> > >
> > > In other words, their additional results relate only to this class of games. Supermodular games are games where the strategies of each player have a lattice structure such that the incentives of one agent to choose a higher strategy (in their lattice) increases as other players switch to higher strategies (on their lattices). See more details here:
> > > https://ocw.mit.edu/courses/electrical-engineering-and-computer-science/6-254-game-theory-with-engineering-applications-spring-2010/lecture-notes/MIT6_254S10_lec07.pdf
> > >
> > > Supermodular games are an interesting class as they are dominance solvable if and only if they have a unique Nash equilibrium (this is clearly not the case for general games, e.g. Rock-Paper-Scissors has a unique Nash but is clearly not dominance solvable). However, this is a *restricted* class of games (see discussion in the bottom of page 2 here: https://pdfs.semanticscholar.org/e889/fcddbe6bb87ee9a0de6a93ef25fa17f366ee.pdf )
> > >
> > > Our results hold for *general games* that are dominance solvable, but only to the specific learning rules we've examined (REINFORCE, IW-MCPI). In contrast, the results you discuss hold for a much wider class of learning rules, but only for the restricted class of supermodular games.
> > >
> > > Again, we hope this helps address your concerns about our results being subsumed by earlier work.

---

> > > > ### Comment · AnonReviewer1 · 2019-11-15
> > > > **Response**
> > > >
> > > > Can you explain why the Proof of Thm 8 doesn't apply to *all* games with an iterated dominance solution? It's a simple induction that says that given the weak condition (A6) the agents will eventually be restricted to playing only best-responses to the initial strategies B(x), and then best responses to those B^2(x), and so on for any B^k.
> > > >
> > > > In iterated dominance solvable games, B^k(x) is the Nash as k --> \infty, right?

---

> > > > > ### Author Response · Authors · 2019-11-15
> > > > > **Theorem 8 relates only to supermodular games.**
> > > > >
> > > > > Milgrom and Roberts's  Theorem 8 statement says that:
> > > > > "Let {x(t)} be an adaptive dynamic process and let x = inf(S) andX = sup (S). Then for every *supermodular* game T..."
> > > > > i.e. the theorem relates only for supermodular games.
> > > > >
> > > > > Given this theorem and a *supermodular game*, one could indeed run through the iterated elimination sequence. However, the theorem only applies for supermodular games (which is again a restricted class of game).
> > > > >
> > > > > You've asked where specifically in the proof of their Theorem 8 Milgrom and Roberts use the fact that the game is supermodular. Their proof of theorem 8 applies their Theorem 5 (or rather, their Lemma 1 which is the key element of Theorem 5). In the proof of Theorem 8, in the last transition (top of page 1270), the authors explicitly state the last transition follows from Lemma 1.
> > > > >
> > > > > Their Lemma 1 again relates only to supermodular games: note that the Lemma is proved using Theorem 1 and Theorem 2, which require a supermodular function f and a lattice (the definition of a supermodular game requires having these). Note that x \wedge y and x \vee y in the definitions of the paper relate to the supermum and infimum of a lattice. Their Lemma 1 leverages the fact that the smallest and largest best response are defined, which requires invoking their Theorem 1 and their Theorem 2, which again relate only to *supermodular* games (the definition of a supermodular game requires having such a lattice structure and supermodular function).
> > > > >
> > > > > In short, Theorem 8 applies only to supermodular games and not general games - the condition A6 is used along their Lemma 1 which relates to supermodular games. Indeed, for the *restricted class* of supermodular games Milgrom and Roberts results are sufficient to show that many learning dynamics converge to the iterated elimination sequence. However, our results hold for *general games* that are dominance solvable, which is a much larger class of games (though our proof is only for specific learning dynamics of REINFORCE and IW-MCPI).

---

### Official Review · AnonReviewer3 · 2019-10-26
**Official Blind Review #3**

**Rating:** 6

**Review:**

The main idea of this paper is to solve multi-agent reinforcement learning problem in dominance solvable games. The paper reviewed general multi-agent reinforcement learning and general norm-form game in game theory. The authors aim to recover multi-agent policies through independent MARL in norm-form dominance-solvable games. The paper states that one of solution concepts of dominance-solvable games is iterated dominance solution, which is different from Nash Equilibrium and may be more suitable under certain scenarios. Furthermore, the paper considers two common RL methods for control and learning policy: REINFORCE and Monte-Carlo policy iteration. The main contribution of the paper is to prove that both REINFORCE in binary action case and Monte-Carlo algorithms find the agents’ policies converging to the iterated dominance solution. The interesting aspect of this paper is that iterated dominance solution based reward scheme can guarantee convergence to the desired agents policies at a cheaper cost in practical principal-agent problems. In appendix, the paper extended its conclusion to Markov games and three possible action cases. To the current status of the paper, I have a few concerns below.

1.	It takes too much space for preliminary work and basic concepts, in Sec 1.1 (preliminary) and Sec 2 (MA-RL and Dominance-Solvable Games).
2.	The notations are inconsistent and unnecessarily complicated. For example, for agent i “its possible actions are the strategies in S_i” (section 2); any action “a \in S_i” (section 2,1); for agent i “for all s_i \in S_l” (Algorithm 1 line 2). It can be consistent to use the same notation to describe the same term. Moreover, “a score per action, x_1, …, x_{m_i}” and “each agent starts with initial logits for x_1, …, x_n”. Formally, the corner mark in the same location should represent the uniform meaning.
3.	Typos: lemma 3.1 proof “g = … (r_{s_h}-r_{s_h})” should be “g = … (r_{s_h}-r_{s_l})”; above section 3.2 “our proof of Theorem 3.1”, should be “Lemma 3.1”.


**Experience Assessment:**

I have published in this field for several years.

**Review Assessment: Checking Correctness Of Derivations And Theory:**

I carefully checked the derivations and theory.

**Review Assessment: Checking Correctness Of Experiments:**

I carefully checked the experiments.

**Review Assessment: Thoroughness In Paper Reading:**

I read the paper thoroughly.

---

> ### Author Response · Authors · 2019-11-14
> **We will condense the definitions and preliminaries sections, and fix the typos. Thanks!**
>
> Thank you for your helpful comments.
>
> We are condensing the the sections on definitions and preliminaries to get to the point more quickly.
> We are also fixing the typos and the notation inconsistencies you noted - much appreciated!

---

### Official Review · AnonReviewer2 · 2019-10-28
**Official Blind Review #2**

**Rating:** 3

**Review:**

This paper studies independent multi-agent reinforcement learning (MARL) in dominance solvable games. The main contribution of this paper is that the authors have proved the convergence to the iterated dominance solution for two RL algorithms: REINFORCE (Section 3.1, binary action case only) and Importance Weighted Monte-Carlo Policy Improvement (IW-MCPI, Section 3.2). Empirical analysis for principal-agent games is demonstrated in Section 4.

The paper is interesting in general, however, I do not think this paper has quite met the (very high) standard of ICLR, due to the following limitations:

1) As the authors have mentioned, the dominance solvable games are quite limited.

2) This paper only has *convergence* results, but does not have *convergence rate* results. In other words, the authors have not proved how fast the agents converge to the iterated dominance solution. Might the authors establish a convergence rate result such as a regret bound?

3) This paper assumes an unrealistic setting in which when one agent learns, the strategies (policies) of all the other agents are fixed. In other words, the agents learn in a round-robin fashion, rather than learn simultaneously. I do not think this setting is realistic in most practical problems.

**Experience Assessment:**

I have read many papers in this area.

**Review Assessment: Checking Correctness Of Derivations And Theory:**

I assessed the sensibility of the derivations and theory.

**Review Assessment: Checking Correctness Of Experiments:**

I assessed the sensibility of the experiments.

**Review Assessment: Thoroughness In Paper Reading:**

I read the paper at least twice and used my best judgement in assessing the paper.

---

> ### Author Response · Authors · 2019-11-14
> **We have added an analysis of convergence rates. Also, we DO cover the case of simultaneous updates.**
>
> Thank you for the comments.
>
> First, regarding item 3, as we wrote in the original submission (see first paragraph of Section 3 on page 4), our results hold for *both* the “round-robin” setting where one agent learns at a time (which we call the serial mode) and for the case where all agents learn simultaneously (which we call the parallel mode). As we discuss there, our analysis is more elaborate, as it covers the parallel mode as well. In other words, we agree the more realistic setting is the one where all agents learn simultaneously, and our results hold for this more realistic case as well. We’ll emphasize this earlier in the paper.
>
> Second, as you suggest in item 2 regarding convergence rates, we have added a discussion of the convergence rate in an appendix (see “Convergence Rates for IW-MCPI"), which we briefly discuss here.
>
> Our convergence result was an asymptotic one, showing that eventually IW-MCPI almost surely converges to an iterated elimination solution. Note that even in a single bandit settings, the number of samples required to discern that one action x yields a better reward than another action y with high probability 1-\delta depends on the difference in rewards r_x - r_y. Similarly, in our case the rate of convergence depends on the game’s payoffs, with the key factor being the degree to which dominated actions are suboptimal, as captured by the g in the proof our Theorem on IW-MCPI convergence.
>
> In our proof g denotes to the gap in rewards between a dominated action i and any dominating action j (see Theorem 3.4, and note rewards are normalized to be in [0,1]). Denote the total number of actions across all the players as S = \sum_i |S_i| (and note this bounds the number of elimination steps). Denote by epsilon the minimal gap between the dominated action and the other actions across all the elimination steps (i.e. in all elimination steps, the difference between the reward of the eliminated action and the reward of other actions is at least epsilon). Then the required number of steps so that IW-MCPI reaches the iterated elimination solution with high probability is:
> O( S / \epsilon^(2 / (1-p)))
> where p is the decay rate of our exploration.
> For instance, setting p=1/2 we get a convergence time of:
>  O(S / epsilon^4).
>
> Finally, regarding 1, as we wrote in the paper, we acknowledge that many games are not dominance solvable and are not covered by our results. We thus emphasized the implications of our work to mechanism design settings, where one can, under some costs or restrictions, design the game as to make it dominance solvable. We will expand the discussion to emphasize the implications of this.
>
> We hope this addresses your concerns regarding the paper.

---

### Official Review · AnonReviewer4 · 2019-10-30
**Official Blind Review #4**

**Rating:** 6

**Review:**

This paper studies reinforcement learning algorithms in a specific subset of multi-agent environments that are 'dominance solvable'. This means that, given an initial set of strategies in the game, if we iteratively remove 'dominated strategies' (those whose utility is strictly less than another strategy independent of the strategies used by other agents), then only one strategy remains for each player. The remaining strategy is called the iterated dominance solution. The paper proves the convergence of certain RL algorithms (REINFORCE in the 2-action case, and importance weighted monte-carlo policy iteration in the multi-action case) for normal-form games. The paper demonstrates the utility of this via mechanism design: in a principal-agent problem where one can design the rewarding scheme given by a 'principal agent' to various (RL) sub-agents, rewarding schemes motivated by iterated dominance guarantees the best solution for the principal agent, whereas schemes motivated by Nash equilibria do not.

The paper is quite well-written and understandable. To my knowledge, the idea is novel and has not yet been explored in the RL literature (UPDATE: based on Reviewer #1's review, this may not be the case. I'll wait to hear the author response to this). I did not check the proofs thoroughly. However, the experiments in the principal-agent problem make sense, and it's interesting to see that iterated dominance reward schemes results in good performance for the principal agent. I appreciate that, while the main results in the paper are limited to normal-form games (which are quite restricted), there are empirical results in the appendix showing the extension to Markov games with multiple timesteps, suggesting that the applicability of iterated dominance reward schemes extend beyond the simple two-action case, where no temporally extended decisions need to be made. Even so, the Markov game considered is fairly simplistic.

My personal curiosity about this paper revolves around scaling to real-world applications. This is not really discussed in the paper; the conclusion talks about directions for future work, for example expanding the number of RL algorithms where convergence can be proven, or producing complexity bounds for convergence. What I want to know is: what sorts of games can we compute the iterated dominance reward schemes for? How can this be applied when the space of policies becomes too large to be enumerated (and thus determining whether a policy is strictly dominated becomes impossible)? I don't expect this paper to solve these issues, but it would be nice to have a discussion of them.

Overall, I'd say this paper is interesting to the multi-agent RL community and I could imagine others building off of this work, so I err on the side of acceptance.


Small fixes:
- Our proof of Theorem 3.1 -> Theorem 3.2
- I'd recommend extending the captions of figures 6-8 and 9-11 in the Appendix.
- Close bracket in Section 6.3 title

**Experience Assessment:**

I have read many papers in this area.

**Review Assessment: Checking Correctness Of Derivations And Theory:**

I did not assess the derivations or theory.

**Review Assessment: Checking Correctness Of Experiments:**

I assessed the sensibility of the experiments.

**Review Assessment: Thoroughness In Paper Reading:**

I made a quick assessment of this paper.

---

> ### Author Response · Authors · 2019-11-14
> **See comments to reviewer 1 regarding novelty. We will discuss scaling to real world applications.**
>
> Thank you for the helpful comments.
>
> First, regarding novelty: we point out in response to Reviewer 1 that the previous results they noted relate to Fictitious Play and Replicator Dynamics, which are fundamentally different from the RL algorithms we consider. Please see the full comments in our response to Reviewer 1. As we point out, our paper investigates very basic RL methods that are the foundation of many popular RL agents, and we are not aware of any existing work that shows convergence of these methods to the iterated dominance solution. We added a section on “Comparison of Our Results and Existing Results on the Convergence of Other Algorithms to the Iterated Dominance Solution”.
>
> Regarding scaling to real-world applications: we wholeheartedly agree this is an important issue! Indeed, there are known polynomial algorithms for computing strict iterated dominance solutions (see, e.g. Conitzer and Sandholm, Complexity of iterated dominance, 2005, who also note that things are trickier for *weak* iterated dominance). However, these are based on a normal-form representation of a game. Games with multiple timesteps (extensive form) can be translated into normal form, but with the size growing very quickly in the number of timesteps. This means that for practical applications, we might only be able to show convergence for restricted classes of games / environments. We are adding a discussion of this in the paper, and expanding the discussion on mechanism design, where we may want to *design* a game so as to guarantee it has an iterated dominance solution.
>
> We will of course fix the typos you noted and extend the captions in the Appendix - much appreciated!

---

### Decision · Program_Chairs · 2019-12-19

**Decision:**

Reject

**Comment:**

The paper proofs that reinforcement learning (using two different algorithms) converge to iterative dominance solutions for a class of multi-player games (dominance solvable games).

There was a lively discussion around the paper. However, two of the reviewers remain unconvinced of the novelty of the approach,  pointing to [1] and [2], with [1] only pertaining to supermodular games. The exact contribution over such existing results is currently not addressed in the manuscript.  There were also concerns about the scaling and applicability of the results, as dominance solvable games are limited.

[1] http://www.parisschoolofeconomics.eu/docs/guesnerie-roger/milgromroberts90.pdf
[2] Friedman, James W., and Claudio Mezzetti. "Learning in games by random sampling." Journal of Economic Theory 98.1 (2001): 55-84.